# AUTOMATED BENCHMARK GENERATION FOR REPOSITORY-LEVEL CODE INPUT SYNTHESIS VIA COVERAGE-GUIDED FUZZING

## ABSTRACT

Evaluating the capabilities of large language models (LLMs) on practical, repository-level testing tasks is crucial for their effective application in software engineering. Many existing benchmarks rely on human-authored data such as issues, patches, and unit tests, which can limit scalability and introduce risks of solution leakage from training corpora. We introduce TTG-GEN, an automated framework for generating targeted test-input generation (TTG) problems from real-world codebases, in which LLMs are tasked with synthesizing input byte sequences to execute specific, designated code locations. These problems are representative of tasks performed by software engineers during debugging and are designed to probe an LLM's understanding of complex control and data flow in real-world scenarios. TTG-GEN leverages coverage-guided fuzzing (CGF) to identify reachable yet non-trivial target locations that require structure-aware inputs to cover. By automatically generating TTG problems, TTG-GEN offers a practical, scalable, and continuously updatable framework with a low risk of direct solution leakage, suited for evaluating repository-level code comprehension. Using TTG-GEN, we construct TTG-BENCH-LITE, a benchmark of 500 such problems derived from 16 foundational C/C++ software projects. Our evaluation across retrieval-based and agent-based settings shows that even the most capable LLMs solve only 15% of these problems on their first attempt. This indicates that comprehending and manipulating program behavior at the repository level remains a significant hurdle for current models, highlighting a substantial gap between their current abilities and the proficiency required for complex software engineering tasks.

## 1 INTRODUCTION

In recent years, large language models (LLMs) have demonstrated significant improvements in code synthesis, completion, and resolving real-world software issues (Jiang et al., 2024; Zhang et al., 2023), making their application across the entire software engineering (SE) lifecycle, including development, testing, and maintenance, highly promising (Hou et al., 2023; Jin et al., 2024). Moreover, SE tasks often require models to understand and coordinate changes across multiple files and process extensive contexts, providing a rich and sustainable testbed for evaluating the capabilities of LLMs (Hudson et al., 2024; Koohestani et al., 2025). Therefore, the evaluation of LLMs on realistic SE tasks is crucial for both understanding their current abilities and effectively applying them to practical software tasks.

Many benchmarks have been proposed to evaluate LLM capabilities on complex SE tasks, ranging from issue resolution to unit test generation (Gu et al., 2024; Jimenez et al., 2023; Mündler et al., 2024; Xu et al., 2025b). However, these benchmarks exhibit certain limitations: they are either (1) unrepresentative of real-world scenarios, or (2) rely heavily on human-written issues, patches, or unit tests for problem creation and result verification. Consequently, this reliance limits scalability, can lead to benchmark saturation, and introduces a risk of models recalling solutions seen during pre-training (Hudson et al., 2024; Koohestani et al., 2025; Cheng et al., 2025; Dong et al., 2024). Therefore, there is an urgent need for evaluation frameworks that are programmatically generated

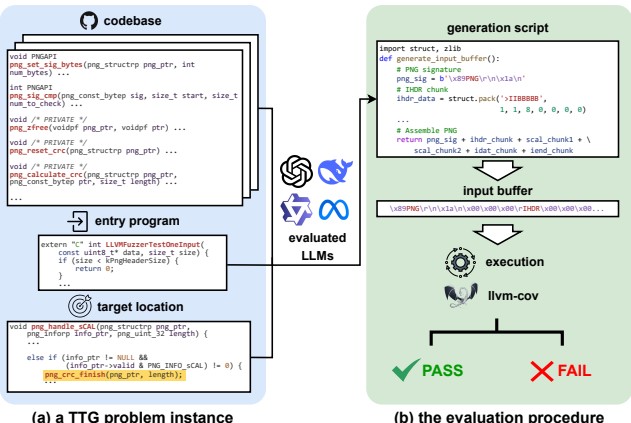

Figure 1: The TTG problem and its evaluation process. (a) A TTG problem instance, comprising a codebase, an entry program, and a target code location. (b) The evaluation procedure, where the LLM-synthesized input is executed and the coverage is analyzed to determine if the target is reached.

for scalability, carry a low risk of solution leakage, and are grounded in realistic software testing scenarios.

The targeted test-input generation (TTG) problem (see Figure 1(a) for an instance), an important task in software testing that involves creating inputs to execute specific code locations (Böhme et al., 2017; Wang et al., 2020), is well-suited for evaluating LLMs. As a benchmark, TTG allows for scalable problem generation with a low risk of solution leakage, and success is unambiguously verified through execution (Pezzè & Young, 2008; Clarke et al., 2018). A key difficulty, however, is selecting suitable targets, since random sampling yields many unreachable or trivial locations that do not adequately test a model's capabilities (Babić et al., 2019; Metzman et al., 2021; Hamlet, 1994; Arcuri et al., 2011). To overcome this, we leverage coverage-guided fuzzing (CGF) to identify locations that are demonstrably reachable yet non-trivial to cover (Zhu et al., 2022; Manès et al., 2019; Li et al., 2018; Mallissery & Wu, 2023). Our key insight is that targets discovered in later CGF stages require more structure-aware inputs. We present TTG-GEN, a method that uses CGF and llvm-cov (LLVM, 2025) to systematically select these newly covered locations, ensuring the generation of high-quality problems for assessing system-level code comprehension (Section 3.1.3).

We constructed TTG-BENCH-LITE, a benchmark designed to evaluate LLMs' code reasoning abilities in real-world scenarios using TTG problems. For this purpose, 16 real-world open-source C/C++ projects from a wide range of domains were selected as inputs to TTG-GEN. From the resulting set of target locations, 500 were randomly selected to form the benchmark. We evaluated several commonly used LLMs on TTG-BENCH-LITE. To handle the large scale of the code repositories, two standard settings (retrieval-based and agent-based) were employed. The result shows that the best-performing model achieved a pass@1 rate of 14.56%, highlighting the challenge presented by TTG-GEN to LLMs. In some cases, LLMs performed similarly to human experts, but in others, they made mistakes, showing their deficiencies in precise reasoning, plausible reasoning and knowledge application, indicating that there is a large room for improvement in their reasoning capabilities. We empirically investigated these failures and discussed the reasons in Section 4.2.2.

In summary, the contribution of this paper is threefold:

- We propose TTG-GEN, a method for generating TTG problems for LLM benchmarking using CGF. TTG-GEN can generates non-trivial problems at scale, offering a continuously updatable evaluation framework with a low risk of solution leakage for assessing repository-level code comprehension.

- We construct TTG-BENCH-LITE, a practical benchmark for evaluating repository-level input synthesis abilities in LLMs. It consists of 500 TTG problems created by applying TTG-GEN to 16 real-world C/C++ repositories that process well-known file formats, facilitating efficient evaluation.

- We conduct a comprehensive evaluation on TTG-BENCH-LITE with widely used LLMs under various settings. The results show that even the best-performing LLM can only achieve a pass@1 of 14.56%, indicating these tasks are a significant hurdle for current models and highlighting considerable room for improvement in LLMs.

The code for TTG-GEN and TTG-BENCH-LITE, along with the containerized running environment, is currently available in supplementary materials and will be open-sourced after review.

## 2 RELATED WORK

**LLM Benchmarks for Code Reasoning and Testing.** Recently, various benchmarks have been proposed to evaluate the capabilities of LLMs on code-related tasks and in practical software testing scenarios (Hudson et al., 2024; Koohestani et al., 2025). Several focus on function-level code comprehension: CRUX-Eval (Gu et al., 2024) and its multilingual extension CRUX-Eval-X (Xu et al., 2024) assess input/output prediction for synthesized functions, while R-Eval (Chen et al., 2024) targets the inference of execution states. Another line of work evaluates unit test generation for real-world Python code. This includes SWT-bench (Mündler et al., 2024) for creating issue-reproducing tests, TestGen-Eval (Jain et al., 2025) for improving coverage, and CLOVER (Xu et al., 2025a) for meeting specific test objectives. Other benchmarks include Test-Eval (Wang et al., 2024) for generating inputs to cover LeetCode problems and Test-Bench (Zhang et al., 2024a) for generating unit tests in Java.

**Coverage-Guided Fuzzing**. Coverage-guided fuzzing (CGF) is a prominent and highly effective automated technique for generating test inputs, widely applied to test complex, real-world systems (Zhu et al., 2022; Manès et al., 2019; Li et al., 2018). At its core, CGF employs an evolutionary algorithm that operates on a corpus of seed inputs, treating them as a population to be evolved (as shown in Figure 3 (b)). The key to CGF's effectiveness is its feedback mechanism. The fuzzer monitors program execution to determine if the mutated input triggers new code coverage, such as executing previously unreached basic blocks, edges, or paths. If an input is deemed "interesting" for having increased coverage, it is retained and added to the seed pool for subsequent mutation rounds; otherwise, it is discarded. This simple yet powerful feedback loop continuously guides the search toward deeper and more complex program states. This technique has proven highly successful in practice, with widely adopted tools such as AFL (Google, 2015a), LibFuzzer (LLVM, 2015), and Honggfuzz (Google, 2014) making significant contributions to software security by uncovering thousands of vulnerabilities.

## 3 METHOD

### 3.1 THE TTG PROBLEM

#### 3.1.1 TASK FORMULATION

**Problem Definition.** Formally, let $R$ be a code repository, $E$ be an entry program that takes a byte sequence (buffer) $b \in B$ as input, and $L_{target}$ be a specific target code location within $R$. An instance of the targeted test-input generation (TTG) problem is a tuple $P = \langle R, E, L_{target} \rangle$. Let $exec(R, E, b)$ be a function that returns the set of code locations executed when program $E$ is run with input buffer $b$. The objective is to find a byte sequence $b$ such that the target location is executed: $L_{target} \in exec(R, E, b)$.

**Solution Representation.** While the goal is to find the byte sequence $b$, LLMs often exhibit limitations in precise numerical calculations and direct binary data manipulation (Steyvers et al., 2025; Akhtar et al., 2023). Consequently, we do not require the model to output the raw byte sequence directly. Instead, the task is to generate a Python script $S_{gen}$ which, when executed, produces the target byte sequence $b$.

#### 3.1.2 RATIONALE FOR TTG AS AN EVALUATION TASK

The TTG problem is an important task in software testing, reflecting a common scenario faced by developers during testing and debugging (Pezzè & Young, 2008; Clarke et al., 2018). From

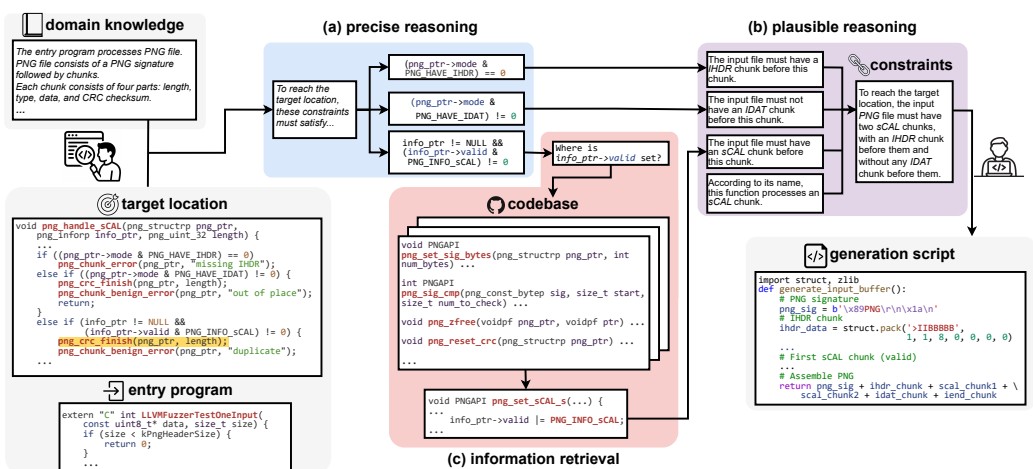

Figure 2: The reasoning process of a human analyst for a TTG problem. This process involves: (a) precise reasoning on code semantics, (b) plausible reasoning with domain knowledge and heuristics, and (c) information retrieval from the codebase.

a practical perspective, it is a core element of software testing, verification, and analysis. While this task is addressable by human experts (Brooks, 1983; Woodfield et al., 1981; Padioleau et al., 2009), traditional automated methods often face significant hurdles like path explosion (Baldoni et al., 2018; Pelánek, 2008). The capabilities of LLMs present a new avenue for tackling these long-standing issues by leveraging domain knowledge and heuristics. From a benchmarking perspective, TTG offers a robust framework for assessing a range of LLM capabilities in repository-level code comprehension, with success being unambiguously verifiable through execution. Furthermore, this approach enables automated generation of new problem instances at scale and carries a low risk of solution leakage, as the required inputs are not typically available in public data.

### 3.1.3 TASK MOTIVATION AND REQUIRED CAPABILITIES

To illustrate the capabilities required to solve such problems, we consider an example from *libpng*, where the goal is to execute a specific line within the *png_handle_sCAL* function (Figure 2).

An expert solves this by leveraging domain-specific knowledge, such as the *PNG* file format specifications, and performing a detailed analysis of the source code to trace control flow. The crucial step is mapping low-level code conditions to high-level properties of the input format, for instance, deducing that the presence and order of specific *PNG* chunks are required. This process of abstraction and constraint satisfaction enables the efficient construction of a valid input. As a benchmark, TTG is designed to assess a similar set of key abilities in LLMs, including: (1) Code Semantic Understanding, to correctly interpret program behavior; (2) Constraint Inference, to determine the precise conditions and execution paths needed to reach a target; (3) Abstract Mapping, to translate low-level code conditions into high-level input format constraints; and (4) Knowledge Application, to use domain-specific information to synthesize a valid input.

### 3.2 THE TTG-GEN METHOD

**Motivation.** While TTG problems are suitable for evaluating LLMs, selecting appropriate targets is critical for a meaningful benchmark. Randomly sampling locations is ineffective for several reasons. First, many locations in large code repositories are simply unreachable by a given entry program (Babić et al., 2019; Metzman et al., 2021). Second, many reachable locations, particularly in input parsing code, can be covered by simple random inputs and thus do not effectively test a model's capabilities (Hamlet, 1994; Arcuri et al., 2011). Third, some targets may be reached by recalling standard file formats rather than by analyzing specific code paths.

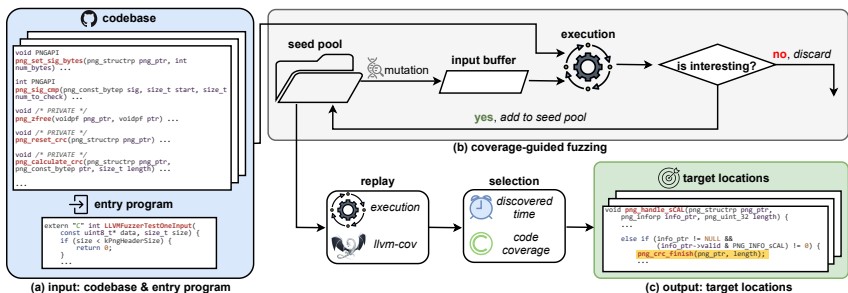

Figure 3: The workflow of TTG-GEN. The process takes (a) a codebase and its entry program as input, then applies (b) coverage-guided fuzzing to it; afterward, the generated seed inputs are replayed to select newly covered locations, ultimately producing (c) a set of curated target locations as output.

**The Method.** To construct a meaningful benchmark, the central task is to identify target locations that are demonstrably reachable yet non-trivial to cover. We find that coverage-guided fuzzing (CGF), a prominent automated software testing technique, is a suitable approach for this selection process. Our key insight is that code locations discovered in the later stages of a CGF campaign serve as ideal targets. The iterative nature of CGF provides three key properties: (1) Feasibility: a location's discovery by the fuzzer confirms it is reachable; (2) Non-triviality: selecting later-discovered locations filters out targets easily covered by simple random inputs; (3) Structural Requirement: reaching these deeper locations often requires generating non-standard or subtly malformed inputs from an initial valid seed pool. This necessitates an analysis of complex code semantics and input structures rather than mere format recall, making the resulting problems well-suited for evaluation.

---

**Algorithm 1** Generating Target Code Locations

---

**Input:** codebase, entryProgram, threshold
**Output:** a set of target locations
seeds, discoveredTime ← CoverageGuideFuzzing(codebase, entryProgram)
seeds ← sortByTime(seeds, discoveredTime)
coveredLocs ← ∅
candidateTargetLocs ← ∅
**for** seed ∈ seeds **do**
   seedCoveredLocs ← getSourceCoverage(codebase, entryProgram, seed)
   newlyCoveredLocs ← seedCoveredLocs \ coveredLocs
   **if** newlyCoveredLocs ≠ ∅ ∧ discoveredTime[seed] ≥ threshold **then**
      targetLoc ← randomChoose(newlyCoveredLocs)
      candidateTargetLocs ← candidateTargetLocs ∪ {targetLoc}
   **end if**
   coveredLocs ← coveredLocs ∪ newlyCoveredLocs
**end for**
**return** candidateTargetLocs

---

The workflow of the proposed TTG-GEN is illustrated in Figure 3. The input to TTG-GEN is a real-world C/C++ repository with an accompanying entry program; the output is a curated set of target locations for constructing TTG problems.

The process begins by applying coverage-guided fuzzing (CGF) to the repository. TTG-GEN employs a hybrid fuzzing approach (Poeplau & Francillon, 2020), which combines a standard fuzzer with a symbolic execution engine to navigate complex path constraints. The initial seed pool is populated with valid input files (e.g., *PNG*, *JPEG*) from established corpora (Metzman et al., 2021), and TTG-GEN utilizes fuzzing dictionaries to guide the fuzzer towards syntactically valid inputs, thereby improving efficiency (Google, 2015b). After a set period of fuzzing, this process yields a corpus of generated inputs, each timestamped with its discovery time.

In the second stage, TTG-GEN analyzes the coverage generated by these inputs. The target program is first compiled with instrumentation flags to generate coverage data. TTG-GEN then replays the generated seed inputs chronologically and use the *llvm-cov* to process the resulting execution profiles (LLVM, 2025). A code location is selected as a candidate target if it is newly covered by a seed whose discovery time exceeds a predefined heuristic threshold (one hour in our experiments). This time-based filtering is designed to select for non-trivial targets that were not discoverable through simple mutations. The overall workflow is summarized in Algorithm 1.

Table 1: Comparison of TTG-GEN and other code reasoning / software testing benchmarks.

| Benchmark | Task | Real-World | Code Scale | Programming Language | Scalable | Low Risk of Solution Leakage | Verifiable |
|---|---|---|---|---|---|---|---|
| CRUX-Eval | Input/Output Pred | ✗ | Function Level | Python | ✓ | ✓ | ✓ |
| CRUX-Eval-X | Input/Output Pred | ✗ | Function Level | Multiple PL | ✓ | ✓ | ✓ |
| R-Eval | Execution State Pred | ✗ | Function Level | Python | ✗ | ✓ | ✓ |
| SWT-Bench | Unit Test | ✓ | Repository Level | Python | ✗ | ✗ | ✗ |
| TestGen-Eval | Unit Test | ✓ | File Level | Python | ✗ | ✗ | ✗ |
| CLOVER | Unit Test | ✓ | Repository Level | Python | ✗ | ✗ | ✗ |
| Test-Eval | Input Generation | ✗ | File Level | Python | ✗ | ✓ | ✓ |
| Test-Bench | Unit Test | ✓ | Class Level | Java | ✗ | ✗ | ✗ |
| TTG-GEN (Ours) | Input Generation / System Test | ✓ | Repository Level | C/C++ | ✓ | ✓ | ✓ |

**Limitations of Existing Benchmarks.** While many benchmarks exist for evaluating LLMs on code-related tasks, they often exhibit shortcomings that limit their utility. Some are constructed from synthesized code snippets or isolated functions, which do not reflect the complexity of real-world software development. Others rely on static, human-authored problem specifications like GitHub issues or existing unit tests for problem creation and verification. This approach limits scalability, can lead to benchmark saturation, and introduces a risk of models recalling solutions seen during pre-training. Furthermore, the scope is often confined to small-scale Python code, leaving system-level testing of large C/C++ applications, which requires analyzing cross-module interactions, largely unexplored. Finally, some benchmarks employ verification methods that can inaccurately assess a model's true capabilities by allowing incorrect solutions to pass existing test suites.

**Advantages of Our Approach.** The proposed TTG-GEN framework addresses these limitations by programmatically generating targeted test-input generation (TTG) problems from real-world C/C++ codebases. The TTG task is grounded in practical software testing and offers a precise, execution-based oracle for verification: a solution is correct if and only if the target location is executed. By leveraging coverage-guided fuzzing (CGF) for target selection, TTG-GEN enables the scalable generation of new problem instances with a low risk of solution leakage. This method ensures that the resulting benchmark is continuously updatable, grounded in realistic system-level contexts, and provides a robust measure of an LLM's ability to comprehend and manipulate complex program behavior (see Table 1).

Table 2: Details of C/C++ repositories used in TTG-BENCH-LITE.

| Repository | Description | Input File Type | #LOC |
|---|---|---|---|
| Bloaty | Program binary size analysis tool | Binary executable files | 984k |
| FreeType2 | Font rendering and processing library | Font files | 441k |
| HarfBuzz | Font shaping engine | Font files | 81k |
| LittleCMS | Color management system | Color profile files | 69k |
| libjpeg-turbo | High-performance JPEG codec library | JPEG images | 87k |
| libpng | PNG image decoding library | PNG images | 138k |
| libxml2 | XML parsing library | XML files | 307k |
| OpenH264 | H.264 video codec library | H264 video files | 131k |
| OpenThread | Implementation of the Thread protocol | IPv6 network packets | 521k |
| RE2 | Efficient regular expression matching library | Regular expressions | 38k |
| libsndfile | Audio file reading/writing library | Audio files | 96k |
| SQLite3 | Lightweight embedded database | SQLite database files | 948k |
| stb | Simple image loading library | Image files | 93k |
| libvorbis | Ogg Vorbis audio codec | Ogg Vorbis audio files | 71k |
| woff2 | WOFF2 font file converter | WOFF2 font files | 57k |
| OpenSSL | Cryptography library with certificate handling | X.509 certificate files | 645k |

**The TTG-BENCH-LITE Benchmark.** Using TTG-GEN, we construct TTG-BENCH-LITE, a benchmark consisting of 500 TTG problems from real-world C/C++ programs to facilitate efficient evaluation. We select 16 widely-used real-world programs that process a wide range of well-known

file formats. The details of the programs are summarized in Table 2. To ensure a fair evaluation, we perform a preliminary check on all participating LLMs. We manually prompt each model to write Python scripts for generating valid files of the specified types and verify the outputs using format-specific parsers. This initial step confirms that the LLMs are capable of writing scripts to produce correctly formatted binary files and possess the necessary domain knowledge of the input formats. We run the fuzzer described for 24 hours, with the discovery time threshold set to one hour. From the resulting target locations returned by TTG-GEN, we randomly select 500 problems to form the problem set.

## 4 EVALUATION

### 4.1 SETTINGS

#### 4.1.1 HANDLING LARGE-SCALE REPOSITORIES

Given a target code location, many parts of the code repository are semantically related to it. However, the large scale of real-world code repositories makes it impractical to fit the entire repository into LLM's context window. To address this, we adopt two standard settings, retrieval-based (Gao et al., 2023; Fan et al., 2024) and agent-based (Jin et al., 2024; Liu et al., 2024b; Yang et al., 2024), as described below. See Appendix A.2 for detailed settings.

**Retrieval-Based Setting.** In this paper, we opt to use BM25 (Robertson et al., 2009) as the retrieval method, following related works in repository-level benchmarking (Jimenez et al., 2023; Mündler et al., 2024). Given the target code location, we concatenate its surrounding code (within 10 lines) along with the signatures of related functions, classes, etc., to form the query string. The source code is split into syntax units (e.g., functions, classes, structs, methods) using Clang (Lattner & Adve, 2004). For syntax units longer than 100 lines, we further split them until every units are less than or equal to 100 lines. Code comments are retained as they are crucial for understanding the semantic of code snippets, as discussed in Section 3.1.2. The query strings and splitted syntax units are tokenized using Clang. We then apply a standard BM25 retrieval approach to retrieve relevant syntax units, ensuring they fits within the context length of 15000 tokens.

**Agent-Based Setting.** Agent-based methods (Jin et al., 2024; Liu et al., 2024b) are commonly used for repository-level software engineering tasks, where LLM agents are equipped with tools to browse, search, and edit code. However, the tools provided to agents in prior works are typically focused on issue resolution or unit test generation tasks (Yang et al., 2024; Zhang et al., 2024b) and may not be as effective for solving the TTG problem. To address this, we design a code browsing tool with functionalities tailored specifically to the needs of the TTG problem, drawing from routines frequently used by human analysts, such as query all code snippets where a given function is called (details see Table 5 in Appendix). The usage of the tool is demonstrated in prompt with concrete examples. Moreover, the agent is allowed to emit multiple queries to the tool in a single session of conversation, helping to reduce the number of conversation rounds to improve efficiency.

#### 4.1.2 LARGE LANGUAGE MODELS

We consider O3-mini (OpenAI, 2025b), GPT-4o-mini (OpenAI, 2025a), DeepSeek-R1 (Guo et al., 2025), DeepSeek-V3 (Liu et al., 2024a), Qwq (Qwen, 2025b), Qwen series (Qwen, 2025a; 2024), Gemma-3 (Team et al., 2025), Mistral-3.1 (Mistral, 2025), Cogito-V1 (Cogito, 2025), Llama-3.3 (Grattafiori et al., 2024), and GLM-4 (GLM et al., 2024) as the underlying LLMs in both retrieval-based and agent-based settings. Among these, GPT-4o-mini, DeepSeek-V3, Gemma-3, Mistral-3.1, Qwen-2.5, Llama-3.3, and GLM-4 are standard LLMs, while O3-mini, DeepSeek-R1, Qwq, R1-Distill-Qwen and GLM-Z1 are reasoning LLMs (Plaat et al., 2024; Li et al., 2025). Qwen-3 and Cogito are hybrid reasoning models. For these models, we set the parameters to the values recommended by the respective model providers. See Appendix A.1 for more details.

### 4.2 RESULTS

For each LLM, the pass@1-5 scores under both retrieval-based and agent-based settings are reported in Table 3. Overall, the performance of different LLMs varies widely, demonstrating that TTG-

Table 3: The pass@1-5 scores (%) of LLMs on TTG-BENCH-LITE.

| Model | Size | Thinking Mode | Reasoning | Retrieval-based Pass@k | | | | | Agent-based Pass@k | | | | |
|---|---|---|---|---|---|---|---|---|---|---|---|---|---|
| | | | | $k=1$ | $k=2$ | $k=3$ | $k=4$ | $k=5$ | $k=1$ | $k=2$ | $k=3$ | $k=4$ | $k=5$ |
| O3-mini | N/A | | ✓ | 12.00 | 15.20 | 16.62 | 17.44 | 18.00 | 11.96 | 15.90 | 18.06 | 19.56 | 20.80 |
| GPT-4o-mini | N/A | | ✗ | 0.56 | 0.84 | 1.06 | 1.24 | 1.40 | 0.56 | 0.72 | 0.88 | 1.04 | 1.20 |
| DeepSeek-R1 | 671B | | ✓ | 14.56 | 18.36 | 20.48 | 21.96 | 23.00 | 14.44 | 19.38 | 21.84 | 23.40 | 24.60 |
| DeepSeek-V3 | 671B | | ✗ | 7.68 | 11.08 | 13.04 | 14.48 | 15.60 | 8.28 | 11.88 | 14.12 | 15.64 | 16.80 |
| Llama-3.3 | 70B | | ✗ | 2.20 | 3.24 | 3.88 | 4.28 | 4.60 | 1.52 | 2.54 | 3.32 | 3.92 | 4.40 |
| Qwq | 32B | | ✓ | 7.20 | 10.78 | 12.84 | 14.20 | 15.20 | 5.76 | 9.78 | 12.62 | 14.68 | 16.20 |
| Qwen-3 | 32B | On | ✓ | 6.16 | 8.84 | 10.52 | 11.64 | 12.40 | 5.88 | 8.32 | 9.84 | 10.96 | 11.80 |
| Qwen-3 | 32B | Off | ✗ | 1.24 | 1.96 | 2.54 | 3.08 | 3.60 | 2.96 | 4.74 | 6.00 | 7.00 | 7.80 |
| Qwen-3 | 30B-A3B | On | ✓ | 3.24 | 4.60 | 5.34 | 5.84 | 6.20 | 2.36 | 3.62 | 4.54 | 5.20 | 5.60 |
| Qwen-3 | 30B-A3B | Off | ✗ | 0.68 | 1.22 | 1.64 | 1.96 | 2.20 | 0.76 | 1.20 | 1.48 | 1.68 | 1.80 |
| Qwen-2.5 | 32B | | ✗ | 1.56 | 2.38 | 2.90 | 3.28 | 3.60 | 2.16 | 3.18 | 3.90 | 4.48 | 5.00 |
| Gemma-3 | 27B | | ✗ | 0.32 | 0.62 | 0.90 | 1.16 | 1.40 | 0.88 | 1.50 | 1.96 | 2.32 | 2.60 |
| Mistral-3.1 | 24B | | ✗ | 0.04 | 0.08 | 0.12 | 0.16 | 0.20 | 0.16 | 0.30 | 0.42 | 0.52 | 0.60 |
| Cogito-V1 | 32B | On | ✓ | 1.52 | 2.72 | 3.68 | 4.48 | 5.20 | 2.48 | 4.26 | 5.50 | 6.32 | 6.80 |
| Cogito-V1 | 32B | Off | ✗ | 1.12 | 1.94 | 2.66 | 3.28 | 3.80 | 1.24 | 2.10 | 2.76 | 3.32 | 3.80 |
| R1-Distill-Qwen | 32B | | ✓ | 2.84 | 4.70 | 6.02 | 6.96 | 7.60 | 1.80 | 2.98 | 3.80 | 4.44 | 5.00 |
| GLM-4 | 32B | | ✗ | 0.96 | 1.80 | 2.54 | 3.20 | 3.80 | 1.52 | 2.66 | 3.54 | 4.24 | 4.80 |
| GLM-Z1 | 32B | | ✓ | 2.24 | 3.36 | 4.14 | 4.80 | 5.40 | 2.20 | 3.36 | 4.18 | 4.84 | 5.40 |

BENCH-LITE has good discriminability, which is crucial for benchmarking. The best-performing model (DeepSeek-R1) achieves a pass@1 score of 14.56% and a pass@5 score of 24.60%, and most median-sized LLMs can achieve only pass@1 scores of 0.04%-7.20% and pass@5 scores of 0.20%-16.20%, indicating that TTG-BENCH-LITE is challenging for state-of-the-art LLMs.

### 4.2.1 OBSERVATIONS

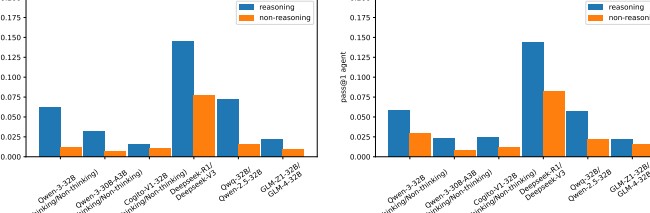

Figure 4: Comparison between reasoning LLMs with their non-reasoning counterparts.

**Comparison of reasoning and non-reasoning models.** To assess the impact of reasoning capabilities gained from reinforcement learning (Plaat et al., 2024; Li et al., 2025) on LLMs' performance in solving the TTG problem, we compare the pass@1 scores of different reasoning LLMs with their non-reasoning counterparts. Six pairs of LLMs are used for this comparison: hybrid reasoning models with the reasoning option on and off (Qwen-3-32B, Qwen-3-30B-A3B, and Cogito-V1), and reasoning versus non-reasoning models that share the same base architecture (DeepSeek-R1/DeepSeek-V3, Qwq-32B/Qwen-2.5-32B, GLM-Z1-32B/GLM-4-32B). Figure 4 compares the pass@1 scores for these model. Across all six pairs, reasoning models outperform their non-reasoning counterparts by a large margin, indicating that the reasoning ability acquired through RL significantly enhances performance in solving TTG problems.

**Comparison of retrieval-based and agent-based settings.** The performance under retrieval-based and agent-based settings varies across different models. The pass@1 scores of various models under both settings are visualized in Figure 5, with reasoning models shown on the left of the dashed line and non-reasoning models on the right. In general, for pass@1, the retrieval-based setting tends to yield better performance for most reasoning models, while the agent-based setting provides superior results for most non-reasoning models. From our manual examinations, we speculate that this is because (1) reasoning models excel at handling long contexts by generating longer chains of thought, which allows them to better reason about complex relationships within the code, and (2) reasoning models are more prone to hallucination, which makes them less effective at using the tools provided in the agent-based setting. Detailed analyses can be found in Appendix D.1.

**Performance of LLMs across different repositories.** The pass@1 scores of top-performing LLMs across 16 C/C++ repositories are shown in Figure 6 in Appendix. The scores vary dramatically, with

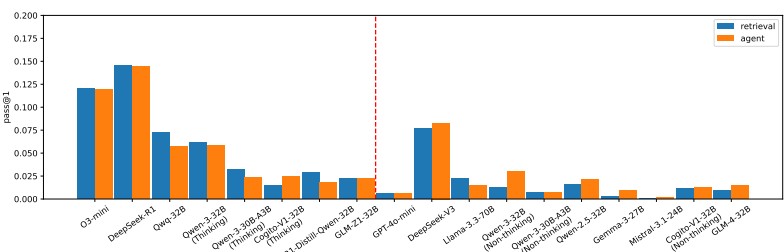

Figure 5: Pass@1 scores of LLMs under different settings.

some repositories achieving near 50%, while others are close to 0%. This variation can likely be attributed to the inherent differences in the complexity of the repositories and their entry programs, and suggests that current LLMs may struggle with reasoning in more complex contexts, where intricate code structures and dependencies are involved.

**Solved problems by different LLMs.** We analyzed the problems solved by different LLMs across all 500 problems in 5 trials, under both retrieval-based and agent-based settings. The result can be found in Appendix D.2, which highlights that, despite differing overall performance, many LLMs are able to solve a unique set of problems that the others cannot. This demonstrates the complementary strengths of each model in addressing various aspects of the TTG problem.

### 4.2.2 CASE STUDIES

We manually examined some of the LLMs' responses to solve the TTG problems. In some cases, LLMs performed well, demonstrating impressive abilities in precise reasoning about path conditions and plausible reasoning about the connection between path conditions and high-level file formats, which are similar to human analysts. However, in many cases, LLMs failed to reach the target location, and we identified several reasons for these failures: First, many LLMs (especially reasoning models) tend to hallucinate and rely on fabricated code snippets for decision-making rather than querying (in the agent-based setting). While these fabricated code snippets may seem plausible, they are not contextually appropriate and not reflective of the real codebase. Second, although LLMs have detailed knowledge about valid target file format specifications (examined by manual prompting), they often struggle to synthesize the correct file format (e.g., a valid *IDAT* chunk in a *PNG* file). Sometimes, they miss key specifications, highlighting the difference between knowledge and their application for LLMs. Third, in some instances, LLMs fail to make plausible connections between low-level conditions and high-level constraints on file formats. As a result, they are unable to generate correct test inputs. Fourth, for complex code reasoning scenarios, where constraints are distributed across a broader context rather than within a single function, LLMs sometimes overlook important constraints. These observations suggest that LLMs still have a large room for improvement and are far from achieving the level of expertise exhibited by human analysts. Detailed analyses and examples can be found in Appendix C.

## 5 CONCLUSION

Evaluating Large Language Models (LLMs) on repository-level code comprehension tasks is crucial for understanding their capabilities and applying them to practical software testing. However, existing benchmarks often rely on static, human-authored data for problem creation and verification, which limits scalability and introduces a risk of solution leakage. In this paper, we present TTG-GEN, an automated method for generating targeted test-input generation (TTG) problems representative of scenarios faced by developers. By utilizing Coverage-Guided Fuzzing (CGF), TTG-GEN generates non-trivial and reachable TTG problems from real-world C/C++ repositories, ensuring scalability with a low risk of solution leakage. We applied TTG-GEN to 16 real-world C/C++ packages to construct TTG-BENCH-LITE, a dataset of 500 problems. Our evaluation shows that even the best-performing LLMs can solve only a small fraction of these problems, indicating that system-level code comprehension remains a significant hurdle for current models.

## 6 REPRODUCIBILITY STATEMENT

In the supplementary material, we provide the code of TTG-GEN and the TTG-BENCH-LITE dataset, with a *README* file for illustrating their usage and how to reproduce the results. We have uploaded the supplementary material to the submission site (OpenReview). We believe it can guarantee the reproducibility of our experiments.

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

Table 4: The LLMs used for evaluation.

| Model | Type | Description |
|---|---|---|
| O3-mini | Reasoning-focused | Specialized in complex reasoning tasks, optimized for logical inference. |
| GPT-4o-mini | General-purpose | A smaller variant of GPT-4o, balancing performance and efficiency. |
| DeepSeek-R1 | Reasoning-focused | Designed for enhanced reasoning capabilities, particularly in structured tasks. |
| DeepSeek-V3 | General-purpose | A versatile model with broad applicability across NLP tasks. |
| Qwq | Reasoning-focused | Optimized for step-by-step reasoning and problem-solving. |
| Qwen-3 | Hybrid | Combines general language understanding with specialized reasoning abilities. |
| Qwen-2.5 | General-purpose | A comprehensive series LLMs designed to meet diverse needs. |
| Gemma-3 | General-purpose | A compact yet powerful open-weight model by Google DeepMind. |
| Mistral-3.1 | General-purpose | Efficient and high-performing, suitable for diverse NLP applications. |
| Cogito-V1 | Hybrid | Integrates general language modeling with structured reasoning mechanisms. |
| Llama-3.3 | General-purpose | Meta's open-weight model, optimized for scalability and task generalization. |
| GLM-4 | General-purpose | Supporting multilingual understanding and generation, suitable for a wide range of NLP tasks. |
| GLM-Z1 | Reasoning-focused | Specializes in advanced decision-making and complex reasoning tasks. |

# A  DETAILED EVALUATION SETTINGS

## A.1  LARGE LANGUAGE MODELS

We list the LLMs used for evaluation in Table 4. For closed-source LLMs, we run them by calling APIs provided by the model developers. For open-source LLMs, we deploy them locally on an Ubuntu 22.04.2 LTS machine equipped with Intel Xeon Gold 6338 processors and NVIDIA H100 PCIe GPUs.

## A.2  RETRIEVAL-BASED SETTING

### A.2.1  RETRIVING WITH BM25

To facilitate retrieval using BM25, we divide the source code related to the target location into chunks. We only consider the C/C++ source files that participate in the compilation of the target fuzz driver. For programming languages like C/C++, it is natural to split source code into syntax units for retrieval to preserve semantic integrity. To achieve this, we use Clang to parse the source code into abstract syntax trees (ASTs), where subtrees correspond to syntax units. Initially, we split the code into high-level syntax units (e.g., function declarations, class declarations, method declarations, struct declarations, enum declarations, etc.). For syntax units containing more than 100 lines, we further split them until each unit contains fewer than or equal to 100 lines of code. We then extract the surrounding code of the target location, which is limited to 10 lines or fewer. Next, we extract related function signatures, class definitions, etc., and concatenate them with the surrounding code and the signature of the entry function LLVMFuzzerTestOneInput to form the query string. The tokenization process is carried out using standard C/C++ lexical analysis. We retrieve code chunks based on the query string, calculate their BM25 scores, and sort them according to the scores. Finally, we select as many of the top-scoring chunks as possible, as long as they fit within a 15,000-token limit.

### A.2.2  PROMPT TEMPLATES

We list the prompt templates used in this section. We adopt a one-shot prompt strategy to facilitate effective interactions with the LLMs.

**System Prompt.** The system prompt consists of an objective description and an exemplary interaction. The objective description is shown in Listing A.2.2, and the exemplary interaction is shown in Listing A.2.2.

Listing 1: Objective description in retrieval-based setting

```
You are a software analysis expert. Your goal is to analyze source code to generate input
↪  byte sequences for C/C++ programs, allowing the program to execute and reach specific
↪  code locations when provided with these sequences.
#### Objective Description
```

```
Note that the program may undergo a series of processing and checks before reaching the
↪  specified line while handling the input byte sequence. You need to ensure that all these
↪  checks pass smoothly.
The input byte sequence for the program will be passed to the data parameter of the
↪  function:
```C
int LLVMFuzzerTestOneInput(const uint8_t* data, size_t size)
```
Here, size represents the length of the input bytes. You need to ensure that after reading
↪  the input from the LLVMFuzzerTestOneInput function, the program can execute and reach
↪  the specified code line in the target function. Therefore, you may need to analyze the
↪  call chain from LLVMFuzzerTestOneInput to the target function.
Note: Do not make any assumptions, do not use "possibly"; all information must be derived
↪  from known background knowledge or obtained through code analysis.

##### Format for Returning Input Sequence
If you believe that you can find the input byte sequence without further queries, please
↪  write a Python function named generate_buf. When this function is called, it will
↪  randomly generate a byte sequence that meets the requirements.
You can call external Python packages within this function to perform specific numerical
↪  calculations. Please include the code for this function and any relevant auxiliary
↪  functions in the code field of the response JSON, and place the call chain from
↪  LLVMFuzzerTestOneInput to the target function in the call_chain field.
For example:
```json
{
    "code": "import random\ndef generate_buf():\n    frag_num = 1\n    frag_total = 1\n
    ↪  session_id = random.randint(0, 99999)\n    payload = b'payload'\n    data_part =
    ↪  f\"AAAAA,{frag_num},{frag_total},{session_id},{payload}\"\n    full_sentence =
    ↪  f\"!{data_part}*\\u0007\"\n    return full_sentence.encode('ascii')",
    "call_chain": ["LLVMFuzzerTestOneInput", "func1", "func2", "target_func"]
}
```
```

Listing 2: The exemplary interaction in system prompt

```
#### Session Example
An example of a session process is as follows:
User:
The program point I need to reach is the first line of the following code:
```C
        return 1;
```
This code is located in the function func, and the expanded version of this function is as
↪  follows:
```C
int func(uint8_t ch) {
    if (ch == 'A'){
        return 1;
    }
    return 0;
}
```

The related code snippets is as follows:
```C
int func(uint8_t ch) {
    if (ch == 'A'){
        return 1;
    }
    return 0;
}
int handle_file(const uint8_t* data, size_t size) {
    if (size > 0){
        return func(data[0]);
    }
    else {
        return 0;
    }
```

```
}
int LLVMFuzzerTestOneInput(const uint8_t* data, size_t size) {
    handle_file(data, size);
}
```

Assistant:
To reach this branch, the variable ch needs to equal 'A'.
By analyzing the code, the variable ch comes from the first element of the data array, and
↪  we must ensure that size > 0.
The current function call chain to reach the target code is:
LLVMFuzzerTestOneInput -> handle_file -> func.
In this chain:
LLVMFuzzerTestOneInput calls handle_file via
```c
    handle_file(data, size);
```

handle_file then calls func via
```c
    if (size > 0){
        return func(data[0]);
    }
```

In func, it reaches the target line through
```c
    if (ch == 'A'){
        return 1;
    }
```
and all the if conditions must pass.

From the analysis, the variable ch in the func function comes from the first element of the
↪  data in the handle_file function. The data and size in handle_file are directly passed
↪  from LLVMFuzzerTestOneInput.
Thus, I can conclude that the constraints to reach the target code are:
The length of the input byte sequence must be > 0.
The first element of the input byte sequence must be 'A'.
The input string b'A' meets these requirements.
Therefore, the final answer is:
```json
{
    "code": "def generate_buf():\n    return b'A'\n",
    "call_chain": ["LLVMFuzzerTestOneInput", "handle_file", "func"]
}
```
```

**User Prompt.** The user prompt template is shown in Listing A.2.2, where "code" refers to the 5 lines surrounding the target location, "surrounding_code" refers to the 10 lines surrounding the target location, and "context" refers to the related code snippets retrieved using BM25.

```
Listing 3: User prompt template in retrieval-based setting

The point I need to reach is the first line of the following code:
```c
{code}
```
It is only necessary to execute up to the first line of this code, without considering its
↪  content and execution context.
To facilitate locating this code, below is a snippet of the surrounding code:
```c
{surround_code}
```
The context code is as follows:
```c
{context}
```
This code is in C/C++ package {prog_package}.
```

Table 5: The specialized tool provided in the agent-based setting.

| Input | Description |
|---|---|
| Function name | Query the code of the given function |
| Class name, Method name | Query the code of the given class method |
| Function name | Query all code snippets where the function is called |
| Class name, Method name | Query all code snippets where the method is called |
| Global variable name | Query the definition of the global variable |
| Global variable name | Query all code snippets that access the global variable |
| Class/struct member variable | Query all code snippets that access the member variable |
| Class/struct/enum type name | Query the definition of the class/struct/enum type |
| Macro name | Query the macro definition |

## A.3 AGENT-BASED SETTING

### A.3.1 TOOL SPECIFICATION AND IMPLEMENTATION

The tools provided in related SE agent works are primarily designed for tasks such as issue resolution and unit test generation, which are not well-suited for TTG problems and often incur a significant cost in interactions between LLMs. To address this, we design a code browser with functionalities tailored specifically for TTG problems, drawing inspiration from routines commonly used by human analysts. The functionalities of the code browser are shown in Table 5. We demonstrate the usage of the code browser through prompts with concrete examples, utilizing a multi-round conversation. In this setup, LLMs query the code browser to browse relevant code snippets, which enhances their ability to solve TTG tasks.

### A.3.2 PROMPT TEMPLATES

We list the prompt templates used in this section. We adopt a one-shot prompt strategy to facilitate effective interaction with LLMs.

**System Prompt.** The system prompt consists of the objective description, tool specification, and an exemplary interaction. The objective description is shown in Listing A.3.2. The tool specification is shown in Listing A.3.2. The exemplary interaction is shown in Listing A.3.2.

```
Listing 4: Objective description in agent-based setting

You are a software analysis expert. Your overall goal is to analyze source code to generate
↪   input byte sequences for C/C++ programs, allowing the program to execute and reach
↪   specific code locations when provided with these sequences.
In a single session, you need to make one of the following two choices:
1. If the information is insufficient, return the information you need to inquire about.
2. If the information is sufficient to make a decision, return the input sequence.

Please note that you should choose option 2 only when you are certain that the information
↪   is adequate and that the input sequence can be completely determined. Otherwise, you
↪   should choose option 1 to obtain more relevant information.

#### Objective Description
Note that the program may undergo a series of processing and checks before reaching the
↪   specified line while handling the input byte sequence. You need to ensure that all these
↪   checks pass smoothly.
The input byte sequence for the program will be passed to the data parameter of the
↪   function:
```C
int LLVMFuzzerTestOneInput(const uint8_t* data, size_t size)
```
Here, size represents the length of the input bytes. You need to ensure that after reading
↪   the input from the LLVMFuzzerTestOneInput function, the program can execute and reach
↪   the specified code line in the target function. Therefore, you may need to analyze the
↪   call chain from LLVMFuzzerTestOneInput to the target function.
```

```
Be aware that you must perform program analysis using known information (such as call chain
↪  analysis, path analysis through if conditions, etc.) to guarantee that the provided
↪  input allows execution to reach the specified code line in the target function from the
↪  LLVMFuzzerTestOneInput function. If this cannot be assured, you should use the code
↪  browser to acquire more information until you can guarantee this.

You have the following choices:
1. When the context or function call information provided is insufficient to generate a
↪  suitable byte sequence, you can use the code browser to browse related code snippets to
↪  obtain more context information or function call information. The user will return the
↪  code snippets you request, and you can analyze the code based on this information to
↪  make further decisions. If these new code snippets are still insufficient to generate
↪  the required sequence, you can initiate another request. Please do not assume function
↪  or structure content or input processing conditions.
2. If you believe the information provided is sufficient, you can give the answer directly.
↪  In your final answer, you should explain the call chain from the LLVMFuzzerTestOneInput
↪  function to the target function, where each call relationship should be confirmed based
↪  on the queried code.

#### Format Description
Your response needs to be provided in JSON format. The specific format for the two choices
↪  is described as follows:

##### Format for Returning Input Sequence
If you believe that you can find the input byte sequence without further queries, please
↪  write a Python function named generate_buf. When this function is called, it will
↪  randomly generate a byte sequence that meets the requirements.
You can call external Python packages within this function to perform specific numerical
↪  calculations. Please include the code for this function and any relevant auxiliary
↪  functions in the code field of the response JSON, and place the call chain from
↪  LLVMFuzzerTestOneInput to the target function in the call_chain field.
For example:
```json
{
    "code": "import random\ndef generate_buf():\n    frag_num = 1\n    frag_total = 1\n
    ↪  session_id = random.randint(0, 99999)\n    payload = b'payload'\n    data_part =
    ↪  f\"AAAAA,{frag_num},{frag_total},{session_id},{payload}\"\n    full_sentence =
    ↪  f\"!{data_part}*\\u0007\"\n    return full_sentence.encode('ascii')",
    "call_chain": ["LLVMFuzzerTestOneInput", "func1", "func2", "target_func"]
}
```
```

**Listing 5: Tool specification in agent-based setting**

```
##### Format for Calling the Code Browser
The code browser supports the following functions:

(1). Provide the function name to query the code of that function.
(2). Provide the function name to query all the code that calls that function.
(3). Provide the global variable name to query all the code that accesses that global
↪  variable.
(4). Provide the class/struct member variable name to query all the code that accesses that
↪  class/struct member variable.
(5). Provide the global variable name to query the definition code of that global variable.
(6). Provide the class/struct/enum type name to query the definition code of that
↪  class/struct/enum type.

When calling the code browser, the response JSON needs to include the following fields:

(1). func_name: A list including all the function names whose code needs to be queried; if
↪  there are none, it should be an empty list.
(2). called_name: A list including all the function names whose calling code needs to be
↪  queried; if there are none, it should be an empty list.
(3). global_used: A list including all the global variable names whose accessing code needs
↪  to be queried; if there are none, it should be an empty list.
(4). class_member_used: A list including all the class/struct member variable names whose
↪  accessing code needs to be queried; if there are none, it should be an empty list.
(5). global_def: A list including all the global variable names whose definition code need
↪  to be queried; if there are none, it should be an empty list.
```

```
(6). type_name: A list including all the class/struct/enum type names whose definitions need
↪  to be queried; if there are none, it should be an empty list.

For example, the following response:
```json
{
    "func_name": ["f", "g"],
    "called_name": ["h"],
    "global_used": ["v1"],
    "class_member_used": ["m1"],
    "global_def": ["v2"],
    "type_name": ["clsA"]
}
```

indicates a request to query:
+ The code for functions f and g
+ All code that calls function h
+ All code that accesses the global variable v1
+ All code that accesses the class/struct member variable m1
+ The definition code of global variable v2
+ The definition code of class/structure/enum type clsA

The user will return the results of the query to you.
Do not repeat your query.
```

Listing 6: The exemplary interaction in system prompt

```
#### Session Example
An example of a session process is as follows:
User:
The program point I need to reach is the first line of the following code:
```C
        return 1;
```
This code is located in the function func, and the expanded version of this function is as
↪  follows:
```C
int func(uint8_t ch) {
    if (ch == 'A'){
        return 1;
    }
    return 0;
}
```

Assistant:
To reach this branch, the variable ch needs to equal 'A'.
I cannot determine how the ch variable is obtained from the input, so I need to query which
↪  functions called func.
```json
{
    "called_name": ["func"]
}
```

User:
Here is the code snippet you requested:
```c
int handle_file(const uint8_t* data, size_t size) {
    if (size > 0){
        return func(data[0]);
    }
    else {
        return 0;
    }
}
```

Assistant:
```

```
By analyzing the code, the variable ch comes from the first element of the data array, and
↪  we must ensure that size > 0.
I cannot determine how the data array and size are obtained from the input, so I need to
↪  query which functions called handle_file.
```json
{
    "called_name": ["handle_file"]
}
```

User:
Here is the code snippet you requested:
```c
int LLVMFuzzerTestOneInput(const uint8_t* data, size_t size) {
    handle_file(data, size);
}
```

Assistant:
By analyzing the code, the current function call chain to reach the target code is:
LLVMFuzzerTestOneInput -> handle_file -> func.
In this chain:
LLVMFuzzerTestOneInput calls handle_file via
```c
    handle_file(data, size);
```

handle_file then calls func via
```c
    if (size > 0){
        return func(data[0]);
    }
```

In func, it reaches the target line through
```c
    if (ch == 'A'){
        return 1;
    }
```
and all the if conditions must pass.

From the analysis, the variable ch in the func function comes from the first element of the
↪  data in the handle_file function. The data and size in handle_file are directly passed
↪  from LLVMFuzzerTestOneInput.
Thus, I can conclude that the constraints to reach the target code are:
The length of the input byte sequence must be > 0.
The first element of the input byte sequence must be 'A'.
The input string b'A' meets these requirements.
Therefore, the final answer is:
```json
{
    "code": "def generate_buf():\n    return b'A'\n",
    "call_chain": ["LLVMFuzzerTestOneInput", "handle_file", "func"]
}
```
```

**User Prompt.** The user prompt is shown in Listing A.3.2, where "code" refers to the 5 lines surrounding the target location, "surrounding_code" is the 10 lines surrounding the target location, and "func_code" is the code of the function queried using the function name.

---

Listing 7: User prompt in agent-based setting

```
The point I need to reach is the first line of the following code:
```c
{code}
```
It is only necessary to execute up to the first line of this code, without considering its
↪  content and execution context.
```

```
To facilitate locating this code, below is a snippet of the surrounding code:
```C
{surround_code}
```
This code is located in the function {func_name}, and code of this function is as follows:
```C
{func_code}
```
This function is in C/C++ package {prog_package}.
```

**Query Result Prompt.** The template used for responding to LLM's queries is shown in Listing A.3.2, in which "code_ret" refers to the code snippets returned by the code browser.

Listing 8: Query result prompt in agent-based setting

```
Here is the code snippet you requested:
```c
{code_ret}
```
```

# B  AN EXEMPLARY CONVERSATION WITH LLM

In this section, we present the responses from LLMs that correctly solve the target problems. These examples demonstrate how the LLMs utilize the provided tools and reasoning capabilities to address the challenges posed by the TTG tasks effectively. Each response showcases the model's understanding of the code, the correct application of reasoning, and the ability to retrieve and integrate relevant information to reach a solution.

## B.1  TARGET PROBLEM

The target code location is at the final branch in the starting part of function *png_handle_sCAL*, where the starting part of this function is shown in Listing B.1.

Listing 9: Starting part of target function

```C
void /* PRIVATE */
png_handle_sCAL(png_structrp png_ptr, png_inforp info_ptr, png_uint_32 length)
{
   png_bytep buffer;
   size_t i;
   int state;

   png_debug(1, "in png_handle_sCAL");

   if ((png_ptr->mode & PNG_HAVE_IHDR) == 0)
      png_chunk_error(png_ptr, "missing IHDR");

   else if ((png_ptr->mode & PNG_HAVE_IDAT) != 0)
   {
      png_crc_finish(png_ptr, length);
      png_chunk_benign_error(png_ptr, "out of place");
      return;
   }

   else if (info_ptr != NULL && (info_ptr->valid & PNG_INFO_sCAL) != 0)
   {
      png_crc_finish(png_ptr, length);
      png_chunk_benign_error(png_ptr, "duplicate");
      return;
   }
   ...
```

## B.2 RESPONSE IN RETRIEVAL-BASED SETTING

The response of the LLM in the retrieval-based setting is shown in Listing B.2.

Listing 10: Response of LLM in retrieval-based setting

```
# Think:
Okay, let's tackle this problem. The user wants to generate an input byte sequence that will
↪  cause the libpng library to execute the specified code block in the function handling
↪  the sCAL chunk. The target code is in the else if condition where the info_ptr's valid
↪  flag has PNG_INFO_sCAL set, leading to the lines png_crc_finish and the error message
↪  "duplicate".

First, I need to figure out how the sCAL chunk is processed in libpng. Looking at the
↪  provided code, the function png_handle_sCAL is responsible for handling sCAL chunks. The
↪  specific code block they want to hit is when the info_ptr already has the sCAL chunk
↪  processed (PNG_INFO_sCAL bit set), which triggers the duplicate error.

So, the conditions to reach that branch are:
1. The info_ptr is not NULL.
2. The info_ptr's valid field has the PNG_INFO_sCAL bit set.
3. The current chunk being processed is another sCAL chunk.

Therefore, to trigger the duplicate error, the input must have two sCAL chunks. The first
↪  one will set the valid bit, and the second one will hit the duplicate condition.

Now, how to structure the PNG data to include two sCAL chunks. Let's recall the PNG file
↪  structure. A PNG starts with the signature, followed by chunks. Each chunk has a length,
↪  type, data, and CRC.

The steps to create such a file:
- Start with the PNG signature (8 bytes: 0x89PNG\r\n\x1a\n).
- Include an IHDR chunk (mandatory first chunk).
- Add the first sCAL chunk.
- Add a second sCAL chunk.
- Possibly add an IEND chunk at the end.

The sCAL chunk structure: The chunk type is 'sCAL'. The data for sCAL must start with a unit
↪  type (1 byte, either 1 or 2), followed by two null-terminated strings representing the
↪  width and height. The minimum length is 4 bytes (unit + at least one byte for each
↪  string, but they need to be null-terminated). For example, the first sCAL chunk could
↪  have unit 1 followed by "1\02\0" (though exact data might vary as long as it meets the
↪  length and format requirements).

Wait, the code for png_handle_sCAL has a check: "else if (info_ptr != NULL &&
↪  (info_ptr->valid & PNG_INFO_sCAL) != 0)". So, after the first sCAL chunk is processed,
↪  the valid bit is set. The second sCAL chunk will trigger the duplicate check.

Therefore, the input data must have two sCAL chunks. The first one must be valid enough to
↪  pass the initial checks (like length >=4, etc.) so that it sets the valid bit. The
↪  second one can be minimal, as long as it's recognized as an sCAL chunk again.

Now, constructing the input:

First, the PNG header:
- Signature: 89 50 4E 47 0D 0A 1A 0A

Then, the IHDR chunk:
- Length: 13 (since IHDR data is 13 bytes)
- Type: 73 68 61 6E (IHDR is 'IHDR', so hex 49 48 44 52)
- Data: 00 00 00 01 00 00 00 01 08 03 (example for 1x1 image, 8 bits, palette)
- CRC: computed for the IHDR data.

Then the first sCAL chunk:
- Length: Let's say 5 bytes (unit type (1) + "1\02\0" would be 1 + 1 + 1 +1 = 4? Wait, the
↪  minimum required is 4 bytes (unit + at least 1 byte each for width and height, but they
↪  need to be null-terminated. So "1\0" for width and "\02\0" might not be right. Let me
↪  think again. The minimum length is 4 bytes as per the code: "else if (length < 4) { ...
↪  invalid }". So unit (1 byte) plus the rest must be at least 3 bytes. For example,
↪  unit=1, then "1\02\0" would be 4 bytes. So length is 4.
```

```
So first sCAL chunk:
– Length: 4 (in big-endian 00 00 00 04)
– Type: 'sCAL' (hex 73 43 41 4C)
– Data: 01 31 00 32 (unit=1, width "1", height "2" as strings, but need null terminators.
↪  Wait, the width and height are ASCII numbers separated by a null. So the data would be
↪  [unit][width string][0][height string][0]. So to have both strings, the minimum would be
↪  unit (1) + "a\0b\0" which is 4 bytes total. So for example, unit 1, then "1\02\0" would
↪  be 4 bytes (0x01 0x31 0x00 0x32). That gives unit=1, width "1", height "2". So that's
↪  acceptable.

CRC for the sCAL data (the 4 bytes) would need to be calculated. But since the input is
↪  generated, maybe the CRC can be faked, but the library will process it. Wait, but the
↪  CRC must be correct for the data and length. Hmm, this complicates things because
↪  generating the correct CRC is necessary for the chunk to be processed. Alternatively,
↪  perhaps the code allows some flexibility, but the user's input must have valid CRCs for
↪  each chunk to reach the desired path.

Alternatively, maybe the fuzzer can handle incorrect CRCs but in this case, since the code
↪  calls png_crc_finish, which verifies the CRC, an incorrect CRC would lead to an error
↪  before the duplicate check. Therefore, to trigger the duplicate check, the first sCAL
↪  chunk must have a valid CRC.

So I need to compute the CRC for the first sCAL chunk's data and then append the CRC.

So the first sCAL chunk:

Data part: 0x01 0x31 0x00 0x32 (unit 1, "1\02")

The CRC is computed over the data plus the chunk type. Wait, the CRC in a PNG chunk is
↪  computed over the chunk type and data. So for the sCAL chunk:

The CRC is computed as:

CRC32 of the 4-byte chunk type (sCAL) plus the data (4 bytes), then the 4-byte CRC is
↪  appended.

Wait, the chunk structure is:

[Length (4 bytes)][Chunk Type (4 bytes)][Data (length bytes)][CRC (4 bytes)]

The CRC is computed over the chunk type and data, then stored in the 4 bytes after data.

Therefore, to compute the CRC for the first sCAL chunk:

Data (including the 4 bytes of data):

The chunk type is 'sCAL' (73 43 41 4C in hex, or 0x73 0x43 0x41 0x4C).

The data is 0x01 0x31 0x00 0x32 (assuming the unit is 1, followed by "1\02").

So the bytes to compute the CRC are the 4 bytes of chunk type plus the 4 bytes of data,
↪  totaling 8 bytes.

Compute the CRC32 of those 8 bytes, then append as 4 bytes.

Then, the second sCAL chunk must be another sCAL chunk. Let's make it minimal. The second
↪  sCAL chunk can have the same data, or just the minimal required.

Second sCAL chunk:

Length: 4 again (same data as first, or even shorter? Wait, the minimum length is 4, so it
↪  can be same as first. But to make sure it's processed, it must have a valid CRC.

So the second sCAL chunk would have the same structure as the first, but with the same data,
↪  leading to same CRC?

Wait, but the data can be different. Let's proceed.

Alternatively, perhaps the second sCAL can have a different data, but must have valid CRC.

So, putting this all together, the input data would be:
```

```
PNG signature (8 bytes) +

IHDR chunk (with length 13, type IHDR, data for 1x1 image, and correct CRC) +

First sCAL chunk (length 4, type sCAL, data as above, correct CRC) +

Second sCAL chunk (same structure as first) +

IEND chunk (mandatory at end, with correct CRC).

Now, the problem is generating all these chunks with correct CRCs. But since the user is to
↪  write a Python function to generate this, perhaps they can compute the CRCs on the fly.

Wait, but in the code example given in the problem's example, the user didn't compute CRCs
↪  but just generated the bytes. However, in reality, the CRCs must be correct for the
↪  chunks to be processed.

Alternatively, maybe the fuzzer's code can handle the CRC, but the input must have the
↪  correct structure. Since the user's code is to generate the input bytes, perhaps the
↪  helper functions can compute CRC32.

In Python, the zlib module has a crc32 function. So, in the generate_buf function, we can
↪  compute the CRCs.

Let me outline the steps for the Python code:

1. Generate the PNG signature.

2. Generate the IHDR chunk:

   - IHDR data: For a minimal 1x1 image, 8 bits, palette color type (4). The IHDR data is 13
   ↪  bytes: width (4 bytes, 0x00 00 00 01), height (4 bytes same), bit depth (1 byte 08),
   ↪  color type (1 byte 03 or 04?), compression (0), filter (0), interlace (0). Wait,
   ↪  palette is color type 3. Let's pick 3 for color type.

   So IHDR data is:

   width: 0x00 00 00 01 (1 pixel)

   height: same

   bit depth: 0x08 (8 bits)

   color type: 0x03 (palette)

   compression: 0x00 (deflate)

   filter: 0x00 (no filter)

   interlace: 0x00 (none)

   So data is: b'\x00\x00\x00\x01\x00\x00\x00\x01\x08\x03\x00\x00\x00'

   The length of the IHDR data is 13, so the length field is 0x00 00 00 0D.

   The CRC is computed over the IHDR type and data. The type is 'IHDR' (0x49 48 44 52).

   So the CRC is computed on the 4 bytes of 'IHDR' plus the 13 data bytes.

   Using zlib.crc32:

   ihdr_data = b'IHDR' + b'\x00\x00\x00\x01\x00\x00\x00\x01\x08\x03\x00\x00\x00'

   crc = zlib.crc32(ihdr_data) & 0xFFFFFFFF

   The CRC for IHDR is known to be 0xed c3 5e 9a (but let's compute it properly in code).

3. First sCAL chunk:

   The sCAL chunk data is [unit][width][0][height][0].
```

```
     Let's choose unit 1 (0x01), width "1" (0x31), height "2" (0x32), so the data is
     ↪  b'\x01\x31\x00\x32'.

     The length is 4, so length bytes are 0x00 00 00 04.

     The type is 'sCAL' (0x73 43 41 4C).

     The CRC is computed over 'sCAL' + data (4 bytes), so total 8 bytes.

     So the chunk's CRC is computed on b'sCAL' + b'\x01\x31\x00\x32'.

4. Second sCAL chunk: same as first, so same data and CRC.

5. IEND chunk: standard, with no data. Its length is 0, type IEND. CRC is 0xAE426082.

Putting this together in code:

The Python function generate_buf() would construct all these chunks, compute the CRCs, and
↪  return the concatenated bytes.

Wait, but the user's code may not need to handle all the CRCs correctly, but the problem
↪  states that the input must reach the target code, so the first sCAL must be processed
↪  and set the valid bit, so the CRC must be correct.

Thus, the code must correctly compute CRCs.

So here's the plan for the code:

Import struct and zlib.

Construct each chunk step by step.

First the signature:

signature = b'\x89PNG\r\n\x1a\n'

Then the IHDR chunk:

def make_chunk(type, data):
    length = len(data)
    crc = zlib.crc32(type.encode() + data) & 0xFFFFFFFF
    return struct.pack('!I', length) + type.encode() + data + struct.pack('!I', crc)

Then, create the IHDR data:

ihdr_data = b'\x00\x00\x00\x01' * 2 + b'\x08\x03\x00\x00\x00'  # width, height, etc.
Wait, let me write it properly:

width = 1
height = 1
bit_depth = 8
color_type = 3  # palette
compression = 0
filter_method = 0
interlace = 0

ihdr_data = (
    width.to_bytes(4, 'big') +
    height.to_bytes(4, 'big') +
    bytes([bit_depth, color_type, compression, filter_method, interlace])
)
# So the data is 13 bytes.

Then the IHDR chunk is made with type 'IHDR' and this data.

Then first sCAL chunk:

scal_data = b'\x01\x31\x00\x32'  # unit 1, "1\02"
scal_chunk = make_chunk('sCAL', scal_data)
```

```
Second sCAL chunk: same as first.

Then IEND chunk:

iend_chunk = make_chunk('IEND', b'')

So putting it all together:

def generate_buf():
    import struct
    import zlib

    def make_chunk(type_str, data):
        length = len(data)
        chunk_type = type_str.encode()
        crc_data = chunk_type + data
        crc = zlib.crc32(crc_data) & 0xFFFFFFFF
        return struct.pack('!I', length) + chunk_type + data + struct.pack('!I', crc)

    # Signature
    buf = b'\x89PNG\r\n\x1a\n'

    # IHDR chunk
    ihdr_data = (
        (1).to_bytes(4, 'big') * 2 +  # width and height
        (8).to_bytes(1, 'big') +      # bit depth
        (3).to_bytes(1, 'big') +      # color type (palette)
        b'\x00\x00\x00'               # compression, filter, interlace
    )
    buf += make_chunk('IHDR', ihdr_data)

    # First sCAL chunk
    scal_data = b'\x01\x31\x00\x32'  # unit 1, width "1", height "2"
    buf += make_chunk('sCAL', scal_data)

    # Second sCAL chunk (same as first)
    buf += make_chunk('sCAL', scal_data)

    # IEND chunk
    buf += make_chunk('IEND', b'')

    return buf
```

Wait, but need to ensure that after the first sCAL chunk is processed, the info_ptr's valid
↪  bit is set. The second sCAL chunk will trigger the duplicate check.

This should work. The code creates two sCAL chunks, which will cause the second one to hit
↪  the duplicate condition.

Now, the call chain from LLVMFuzzerTestOneInput to the target code:

The LLVMFuzzerTestOneInput function is part of the fuzzer's driver, which calls into
↪  libpng's reading functions. The main path would be:

LLVMFuzzerTestOneInput calls into the libpng read functions, which process each chunk. The
↪  sCAL chunks are processed via png_handle_sCAL, which is called when the chunk type is
↪  'sCAL'.

Thus, the call chain would be something like:

LLVMFuzzerTestOneInput -> png_read_png (or similar) -> processes each chunk -> when 'sCAL'
↪  chunk is found, calls png_handle_sCAL -> first sCAL sets valid bit, second sCAL triggers
↪  the duplicate check.

Therefore, the call chain can be listed as:

["LLVMFuzzerTestOneInput", "png_read_png", "png_process_chunk", "png_handle_sCAL"]

But I need to check the code provided. Looking at the code given, the function handling the
↪  chunks is part of the png_read functions. The code has functions like png_handle_sCAL,
↪  which is called when the chunk type is 'sCAL'.
```

In the provided code, the function png_read_data is part of the read process, but the actual
↪  handling of each chunk type is via a switch or some dispatch. Looking at the code,
↪  there's a section where chunks are processed:

Looking at the code, there's a function png_read_chunk that reads the chunk type and
↪  dispatches to the appropriate handler. For example, when the chunk type is 'sCAL', it
↪  calls png_handle_sCAL.

Thus, the call chain from the fuzzer's entry point (LLVMFuzzerTestOneInput) would be:

LLVMFuzzerTestOneInput -> (libpng's read functions) -> png_read_chunk -> png_handle_sCAL
↪  (twice, once for each sCAL chunk)

So the call chain would be:

["LLVMFuzzerTestOneInput", "png_read_png", "png_read_chunk", "png_handle_sCAL"]

But the exact names might be different. Since the code provided includes the functions like
↪  png_handle_sCAL, the call chain is:

LLVMFuzzerTestOneInput calls into the libpng processing, which reads the first sCAL chunk
↪  and calls png_handle_sCAL, which sets the valid bit. Then, when the second sCAL chunk is
↪  read, the same function is called again, leading to the duplicate check.

Thus, the call_chain field should list the path from the fuzzer entry to the target code:

["LLVMFuzzerTestOneInput", "png_process_chunks", "png_handle_sCAL"]

But to be precise, based on the provided code, the code that processes each chunk after
↪  reading the chunk type is likely in a function like process_chunk, which calls the
↪  respective handler. Since the code provided includes the function png_handle_sCAL, the
↪  call chain would be:

LLVMFuzzerTestOneInput -> (the code that reads the PNG data) -> when a 'sCAL' chunk is
↪  encountered, calls png_handle_sCAL. The first call sets the valid bit, the second call
↪  triggers the target code.

Therefore, the call chain to the target line (the duplicate check in png_handle_sCAL) is:

LLVMFuzzerTestOneInput -> (the libpng read loop) -> png_handle_sCAL (second time).

So the call_chain is ["LLVMFuzzerTestOneInput", "png_handle_sCAL"]

Wait, but that's too short. The actual call path would go through more functions. Let me
↪  think again.

The fuzzer's function is the entry point, then the libpng library's functions are called to
↪  read the PNG data. The code for the fuzzer's LLVMFuzzerTestOneInput function is provided
↪  as:

```
extern "C" int LLVMFuzzerTestOneInput(const uint8_t* data, size_t size) {
  if (size < kPngHeaderSize) {
    return 0;
  }

...

    }
  }

  png_read_end(png_handler.png_ptr, png_handler.end_info_ptr);

  PNG_CLEANUP
  return 0;
}
```

The code likely initializes a png_ptr and reads the data via png_read_png or similar. The
↪  actual processing of each chunk is done in the png_read functions. The chunk processing
↪  is handled by functions like png_read_chunk and then the respective handlers (like
↪  png_handle_sCAL).

```
Thus, the call chain is:
LLVMFuzzerTestOneInput -> (initialization and calls to libpng's read functions) -> when the
↪   second sCAL chunk is processed, it calls png_handle_sCAL again, leading to the target
↪   code.
Therefore, the call_chain should list the path from the entry point to the target function.
↪   The target function is png_handle_sCAL, but specifically the part where the duplicate
↪   check is made. So the chain is:

LLVMFuzzerTestOneInput -> ... -> png_handle_sCAL (the second time it is called)

But the call chain is the path taken to reach the target line, which is inside
↪   png_handle_sCAL. So the call chain from the entry point to the function containing the
↪   target code is:

["LLVMFuzzerTestOneInput", "png_read_png", "png_process_chunks", "png_handle_sCAL"]

Alternatively, if the code structure is that the main loop in libpng reads chunks and
↪   dispatches to handlers, then the chain is:
LLVMFuzzerTestOneInput -> png_read_start -> ... -> png_read_chunk -> png_handle_sCAL.

But since the exact code isn't fully provided, I'll assume the minimal chain that includes
↪   the handler function.
The key point is that the code in question is in png_handle_sCAL, so the call chain must
↪   include that function.
Thus, the final answer includes the Python code generating the two sCAL chunks with correct
↪   CRCs, and the call_chain as listed above.

Wait, in the code provided, the function png_handle_sCAL has the code block:

else if (info_ptr != NULL && (info_ptr->valid & PNG_INFO_sCAL) != 0)
{
    png_crc_finish(png_ptr, length);
    png_chunk_benign_error(png_ptr, "duplicate");
    return;
}

So to reach this, the first sCAL chunk must have been processed, setting the valid bit, and
↪   the second sCAL chunk's processing will hit this condition.
The code for generate_buf must create two sCAL chunks. The first one must be valid (so that
↪   it processes and sets the valid bit). The second one can be minimal but must be
↪   recognized as an sCAL chunk.
Therefore, the Python function should generate the structure with two sCAL chunks.

Now, testing the code:
The first sCAL's data is 4 bytes (unit + "1\02"), which meets the minimum length of 4 bytes.
↪   The second sCAL can be the same.
Thus, the code should work.

# Answer:

```json
{
    "code": "import struct\nimport zlib\n\ndef generate_buf():\n    def make_chunk(type_str,
    ↪   data):\n        length = len(data)\n        chunk_type = type_str.encode()\n
    ↪   crc_data = chunk_type + data\n        crc = zlib.crc32(crc_data) & 0xFFFFFFFF\n
    ↪   return struct.pack('!I', length) + chunk_type + data + struct.pack('!I', crc)\n\n
    ↪   # PNG signature\n    buf = b'\\x89PNG\\r\\n\\x1a\\n'\n\n    # IHDR chunk for 1x1
    ↪   palette image\n    ihdr_data = (\n        (1).to_bytes(4, 'big') * 2 +  # width and
    ↪   height\n        b'\\x08\\x03\\x00\\x00\\x00'  # bit depth, color type (palette),
    ↪   compression, filter, interlace\n    )\n    buf += make_chunk('IHDR', ihdr_data)\n\n
    ↪   # First sCAL chunk (unit 1, width '1', height '2')\n    scal_data =
    ↪   b'\\x01\\x31\\x00\\x32'\n    buf += make_chunk('sCAL', scal_data)\n\n    # Second
    ↪   sCAL chunk (duplicate)\n    buf += make_chunk('sCAL', scal_data)\n\n    # IEND
    ↪   chunk\n    buf += make_chunk('IEND', b'')\n    return buf",
    "call_chain": ["LLVMFuzzerTestOneInput", "png_read_png", "png_process_chunks",
    ↪   "png_handle_sCAL"]
}
```
```

## B.3 RESPONSE IN AGENT-BASED SETTING

The response of the LLM in the first round is shown in Listing B.3, and the second response is shown in Listing B.3.

Listing 11: Response of LLM in the first round of agent-based setting

```
# Think:
Okay, I need to generate an input byte sequence that will reach the specific line in the
↪  png_handle_sCAL function where png_crc_finish is called with the "duplicate" error. Let
↪  me start by understanding the code structure of png_handle_sCAL.
The target code is inside the else if condition: else if (info_ptr != NULL &&
↪  (info_ptr->valid & PNG_INFO_sCAL) != 0). So to trigger this, info_ptr must not be NULL,
↪  and the PNG_INFO_sCAL flag must be already set in info_ptr->valid.
Now, the function png_handle_sCAL is part of libpng, which processes PNG chunks. The sCAL
↪  chunk is a special chunk, so the input must be a PNG file containing an sCAL chunk. But
↪  the problem is to get the code to process this chunk and hit the duplicate case.
First, I need to trace the call chain from LLVMFuzzerTestOneInput to png_handle_sCAL. Let's
↪  start by finding out how the fuzzer input is processed. Typically, in libpng fuzzers,
↪  the input is treated as a PNG file. The entry point LLVMFuzzerTestOneInput probably
↪  calls libpng functions to read and parse the PNG data.

So, the first step is to check how LLVMFuzzerTestOneInput is implemented. The user hasn't
↪  provided that code yet. Without knowing how the input is passed to libpng's parsing
↪  functions, I can't determine the path. Therefore, I need to request the code for
↪  LLVMFuzzerTestOneInput.

Once I have that, I can see how the data is processed. Suppose LLVMFuzzerTestOneInput calls
↪  png_read functions, which eventually process chunks. The sCAL chunk handler is
↪  png_handle_sCAL. So, the fuzzer input must include an sCAL chunk.

But even if the input has an sCAL chunk, the code will reach png_handle_sCAL. However, the
↪  target condition is that info_ptr's valid field already has PNG_INFO_sCAL set. That
↪  implies that the info struct must have already processed an sCAL chunk before. So,
↪  perhaps the input needs to contain two sCAL chunks. The first one would set the
↪  PNG_INFO_sCAL flag, and the second one would trigger the duplicate check.

Wait, but how does libpng process multiple chunks? When a PNG file is parsed, each chunk is
↪  processed in order. If the first sCAL chunk is processed, then info_ptr's valid bit is
↪  set. Then, if a second sCAL chunk is encountered, the code would check if the valid bit
↪  is already set, leading to the duplicate error.

So, the input needs to be a PNG file with two sCAL chunks. However, the fuzzer's input is a
↪  byte stream. The LLVMFuzzerTestOneInput function likely passes this data to libpng's
↪  parsing functions. Therefore, the fuzzer's input must be a valid PNG file that includes
↪  two sCAL chunks.

But to construct such a PNG file, I need to know the structure. PNG files start with a
↪  signature, followed by a series of chunks. Each chunk has a length, type, data, and CRC.
↪  So, to create two sCAL chunks, I need to structure the input correctly.

But let's confirm the call chain. The user's target is in png_handle_sCAL, which is called
↪  when the sCAL chunk is processed. So, the input must include an sCAL chunk. The code
↪  path to reach the duplicate error requires that when the sCAL chunk is processed, the
↪  info_ptr already has PNG_INFO_sCAL set.

So, perhaps the input must have two sCAL chunks. The first one is processed normally,
↪  setting the valid flag. The second one triggers the duplicate error.

Therefore, the steps would be:

1. The input starts with the PNG signature.
2. IHDR chunk (required).
3. First sCAL chunk.
4. Second sCAL chunk.
5. Other necessary chunks (like IDAT, IEND).
```

```
1566
1567   But wait, the code in png_handle_sCAL checks if the mode has PNG_HAVE_IDAT. If so, it skips
1568   ↪  processing. So the sCAL chunk must come before the IDAT chunk. In PNG structure, chunks
       ↪  after IDAT are not allowed except for IEND. So the sCAL chunks must be between IHDR and
1569   ↪  IDAT.
1570
1571   Moreover, the code checks if (png_ptr->mode & PNG_HAVE_IDAT) != 0, which means if IDAT has
1572   ↪  been processed, the sCAL chunk is out of place. So the sCAL chunks must be before IDAT.
1573   Therefore, the input must be structured as:
1574
1575   PNG signature → IHDR → sCAL → sCAL → IDAT → IEND.
1576
       But creating such a PNG file requires precise byte crafting. However, the fuzzer's input is
1577   ↪  passed via LLVMFuzzerTestOneInput. So the code for that function must be parsing the
       ↪  input as a PNG file. Let's check how the fuzzer test function is structured.
1578
1579   But since the user hasn't provided the code for LLVMFuzzerTestOneInput, I can't proceed.
1580   ↪  Therefore, my first step is to query the code of LLVMFuzzerTestOneInput to see how data
       ↪  is processed. Additionally, I need to confirm how the libpng functions are called in the
1581   ↪  fuzzer. For example, the fuzzer might initialize png_struct and png_info structs, then
1582   ↪  read the PNG data.
1583
       Once I have the code for LLVMFuzzerTestOneInput, I can see how the data is passed to libpng.
1584   ↪  Let's assume that the function uses png_read functions to parse the input data. Then,
       ↪  the input needs to be a valid PNG file with two sCAL chunks.
1585
1586   Moreover, the info_ptr must not be NULL. In libpng, info_ptr is created by
1587   ↪  png_create_info_struct. So the fuzzer probably creates it, so info_ptr is not NULL.
1588
1589   So, the key constraints are:
1590
       - The input is a valid PNG file with correct signature.
1591   - IHDR chunk is present.
       - Two sCAL chunks after IHDR and before IDAT.
1592   - The first sCAL chunk is valid, so that it sets PNG_INFO_sCAL in info_ptr->valid.
1593   - The second sCAL chunk triggers the duplicate check.
1594
       But to construct the sCAL chunks correctly, each must have valid data. The sCAL chunk's data
1595   ↪  starts with a unit byte (1 or 2), followed by ASCII width and height separated by a null
       ↪  byte.
1596
1597   So, the first sCAL chunk must be correctly formatted so that it is accepted, setting the
1598   ↪  valid flag. Then the second sCAL chunk, when processed, will hit the duplicate
       ↪  condition.
1599
1600   Thus, the input bytes need to form a PNG file with these chunks. But generating such a byte
1601   ↪  sequence requires knowing the exact structure.
1602
       Alternatively, maybe the fuzzer input is not required to be a full PNG file. But given that
1603   ↪  the code checks for PNG_HAVE_IDAT and other mode flags, the input must at least have the
1604   ↪  necessary chunks to pass those checks.
1605
       But without seeing how LLVMFuzzerTestOneInput processes the data, it's impossible to
1606   ↪  proceed. Therefore, I need to request the code for LLVMFuzzerTestOneInput and any
1607   ↪  relevant functions in the call chain leading to png_handle_sCAL.
1608   Possible steps for code queries:
1609
1610   1. Query the code of LLVMFuzzerTestOneInput to see how data is processed.
       2. Determine which functions are called from there. For example, maybe the fuzzer uses
1611   ↪  png_read_png or similar functions.
1612   3. Check if info_ptr is created (png_create_info_struct) and passed.
       4. Check the call chain to see how the sCAL chunk is processed.
1613
1614   Therefore, the initial JSON query should be to get the code for LLVMFuzzerTestOneInput and
1615   ↪  any functions it calls that are part of the processing to handle chunks, especially
       ↪  sCAL.
1616
1617   So, the first JSON query would be:
1618
1619   {
```

```
    "func_name": ["LLVMFuzzerTestOneInput"],
    "called_name": [],
    "global_used": [],
    "class_member_used": [],
    "global_def": [],
    "type_name": []
}
```

Once I have that, I can see how data is passed into the libpng functions. For example, if
↪  LLVMFuzzerTestOneInput initializes png_struct and png_info, then starts reading the
↪  data.

After getting that code, I may need to follow the call chain further. For example, if
↪  LLVMFuzzerTestOneInput calls a function like png_read_png, which in turn processes
↪  chunks, leading to png_handle_sCAL.

But to confirm, I might need to query the code of functions called by
↪  LLVMFuzzerTestOneInput, such as png_read_png, or other functions in the call chain.

Another consideration: the fuzzer might have a custom setup. For example, it might use a
↪  file reader that's passed to libpng, which reads from the input buffer.

Assuming that the code for LLVMFuzzerTestOneInput is something like:

```c
#include <png.h>

int LLVMFuzzerTestOneInput(const uint8_t *data, size_t size) {
    png_structp png = png_create_read_struct(...);
    png_infop info = png_create_info_struct(png);
    // Set up error handling, etc.
    png_set_read_fn(png, &data, user_read_fn);
    png_read_info(png, info);
    // ... process chunks
    png_destroy_read_struct(...);
    return 0;
}
```

In this case, the input data is treated as a PNG file. So the input must be a valid PNG file
↪  with the required chunks.

But to reach png_handle_sCAL, the input must include an sCAL chunk. Moreover, the duplicate
↪  case requires two sCAL chunks. But in PNG, chunks are processed in order. The first sCAL
↪  would set the info_ptr's valid flag, the second would trigger the duplicate error.

Therefore, the input must be a PNG file with two sCAL chunks. However, creating such a PNG
↪  requires precise construction, including valid chunk lengths, types, data, and CRCs.

But generating this manually is complex. However, for fuzzing purposes, perhaps the fuzzer
↪  can be tricked into processing the sCAL chunk twice without having to generate a valid
↪  PNG. But given the code checks (like PNG_HAVE_IHDR, etc.), the input must at least have
↪  the necessary chunks to pass these checks.

Alternatively, maybe the code can be reached by a malformed PNG that still manages to
↪  trigger the sCAL processing twice.

But without knowing how the fuzzer passes data to libpng, it's hard to proceed. Therefore,
↪  the immediate step is to query the code of LLVMFuzzerTestOneInput and any functions it
↪  calls that are part of processing the input data.

Another possibility is that the fuzzer uses a custom IO reader that feeds data to libpng.
↪  The code for LLVMFuzzerTestOneInput might set up the read function to read from the data
↪  buffer. But without seeing that code, I can't determine the exact requirements.

So, the next step is to request the code for LLVMFuzzerTestOneInput. Once I have that, I can
↪  trace the call chain further.

Therefore, the initial query should be:

```json
{
    "func_name": ["LLVMFuzzerTestOneInput"],
```

```json
    "called_name": [],
    "global_used": [],
    "class_member_used": [],
    "global_def": [],
    "type_name": []
}
```

# Answer:
```json
{
    "func_name": ["LLVMFuzzerTestOneInput"],
    "called_name": [],
    "global_used": [],
    "class_member_used": [],
    "global_def": [],
    "type_name": []
}
```

---

### Listing 12: Response of LLM in the second round of agent-based setting

```
# Think:
Okay, let's try to figure out how to generate an input that reaches the target line in
↪  png_handle_sCAL. The user wants the input to cause the program to execute the line where
↪  png_crc_finish is called with the "duplicate" error.

First, looking at the provided code for LLVMFuzzerTestOneInput, it processes the input as a
↪  PNG file. The function checks if the input starts with the PNG header (kPngHeaderSize is
↪  8 bytes, the standard PNG signature). If it does, it proceeds to read the PNG using
↪  libpng functions.

The key steps in LLVMFuzzerTestOneInput are:
1. Check the input size is at least the PNG header size.
2. Verify the PNG signature.
3. Create png_struct and png_info structures.
4. Set up custom read functions to read from the input buffer after the header.
5. Read the PNG info, IHDR chunk, and process the image data.
To reach png_handle_sCAL, the input must include an sCAL chunk. The target code is triggered
↪  when a second sCAL chunk is encountered after the first one has already set the
↪  PNG_INFO_sCAL flag in info_ptr.
So, the input PNG needs to have two sCAL chunks. The first sCAL chunk is processed normally,
↪  setting the valid flag. The second sCAL chunk will trigger the duplicate check.
But how does the PNG structure work? PNG chunks must appear in a certain order. The IHDR
↪  must come first, followed by other chunks like sCAL, then IDAT, and finally IEND.
↪  However, the code in png_handle_sCAL checks if IDAT has been processed (PNG_HAVE_IDAT).
↪  If so, it skips processing the sCAL chunk. Therefore, the sCAL chunks must come before
↪  IDAT.

So, the structure of the input should be:
- PNG signature (8 bytes)
- IHDR chunk
- sCAL chunk (first)
- sCAL chunk (second)
- IDAT chunk(s)
- IEND chunk

Each chunk has a length (4 bytes), type (4 bytes), data, and CRC (4 bytes). For the sCAL
↪  chunks, the data must be valid to pass initial checks. The first sCAL chunk needs to be
↪  correctly formatted so that it's accepted, setting the PNG_INFO_sCAL flag.

Looking at the code for png_handle_sCAL, the sCAL chunk's data must start with a unit byte
↪  (1 or 2), followed by width and height as ASCII strings separated by a null byte. The
↪  png_check_fp_number function checks the format of these strings.

To create a valid first sCAL chunk:
- Length: Let's say the data is "1\01.0\01.0" (unit=1, width=1.0, height=1.0). The actual
↪  length would be 1 (unit) + 3 (width "1.0") + 1 (null) + 3 (height "1.0") = 8 bytes. So
↪  length is 8, stored as 4 bytes in big-endian.
- Type: 'sCAL' (ASCII, 4 bytes)
```

```
- Data: The unit byte followed by "1.0\01.0"
- CRC: Calculated over type and data.

The second sCAL chunk can be minimal, maybe even invalid, since we just need it to trigger
↪   the duplicate check. But its type must be 'sCAL', and the length must be at least 4
↪   (since the code checks length < 4 for some conditions). However, since the code checks
↪   if (info_ptr->valid & PNG_INFO_sCAL) != 0, the second sCAL chunk just needs to exist.

But the code for the first sCAL chunk must pass all checks to set the valid flag. Let's
↪   outline the steps for the first sCAL chunk:

1. The chunk's length is >=4 (the code checks else if (length <4) ...).
2. The unit is 1 or 2.
3. The width and height are valid floating-point numbers in ASCII, separated by a null byte.
4. The CRC is correct.

So, the first sCAL chunk must be correctly formatted. The second sCAL chunk just needs to be
↪   present, but since the code checks the valid flag, its data may not matter as long as
↪   the chunk is processed.

Constructing such a PNG file requires precise byte crafting. Let's outline the structure:

PNG signature (8 bytes): \x89PNG\r\n\x1a\n

IHDR chunk (25 bytes total):
- Length: 13 (00 00 00 0D)
- Type: IHDR (49 48 44 52)
- Data: width (4 bytes), height (4 bytes), bit depth (1), color type (1), compression (1),
↪   filter (1), interlace (1)
- CRC: computed over type and data.

Then the first sCAL chunk:
- Length: 8 (00 00 00 08)
- Type: sCAL (73 43 41 4C)
- Data: unit (1, 0x01), width "1.0" (31 2E 30), null (00), height "1.0" (31 2E 30)
- CRC: calculated.

Second sCAL chunk:
- Length: 4 (00 00 00 04)
- Type: sCAL (73 43 41 4C)
- Data: unit (1, 0x01) followed by three arbitrary bytes (since length is 4, but the code
↪   requires more, but perhaps this will trigger the error early)
- CRC: doesn't matter as long as the chunk is processed.

Then IDAT chunk(s) and IEND.

However, the code in png_handle_sCAL for the first sCAL chunk must pass the checks. The
↪   first sCAL chunk's data must be valid. The code reads the entire chunk data, then checks
↪   the unit, then the width and height.

For the first sCAL chunk:
- buffer[0] must be 1 or 2.
- The rest of the data must have width and height as valid numbers.

In the code, after reading the data, it checks:
if (buffer[0] != 1 && buffer[0] != 2) → error.
Then, check the numbers:
i starts at 1. png_check_fp_number checks the width. Then there must be a null, then the
↪   height.

So for the first sCAL chunk, data should be:
unit (1) followed by "1.0\0" and "1.0", totaling 1 + 3 + 1 + 3 = 8 bytes. So length is 8.

The second sCAL chunk can have length 4, but according to the code:
else if (length < 4) → invalid. So length must be >=4. So length 4 is okay, but data must be
↪   at least 4 bytes. The code will read 4 bytes, then check buffer[0], but since the
↪   info_ptr already has PNG_INFO_sCAL, it will trigger the duplicate error.

But wait, the code for the duplicate check is:
else if (info_ptr != NULL && (info_ptr->valid & PNG_INFO_sCAL) != 0)
```

```
So when processing the second sCAL chunk, since info_ptr->valid has PNG_INFO_sCAL set, it
↪  will enter this condition, call png_crc_finish, and return. So regardless of the data in
↪  the second sCAL chunk, as long as the chunk is processed and the valid flag is set, it
↪  will trigger the target line.

Therefore, the input must have two sCAL chunks. The first must be valid to set the flag, the
↪  second can be minimal but valid enough to be processed (length >=4, correct CRC).

Now, constructing such a PNG:

1. PNG signature (8 bytes).
2. IHDR chunk (25 bytes).
3. First sCAL chunk (8 + 4 (type) + 8 (data) + 4 (CRC) = 24 bytes).
4. Second sCAL chunk (4 + 4 + 4 + 4 = 16 bytes, with data length 4, but valid CRC).
5. IDAT chunk (minimal, perhaps 1 byte data).
6. IEND chunk.

But generating the correct CRC values is essential. The CRC is computed over the chunk type
↪  and data. For the first sCAL chunk, the type is 'sCAL' and the data is as above. For the
↪  second sCAL chunk, the type is 'sCAL' and data is 4 bytes (e.g., 0x01 0x00 0x00 0x00).
↪  However, the CRC must be correct, otherwise libpng will ignore the chunk due to CRC
↪  errors.

But in the code provided, png_set_crc_action is called with PNG_CRC_QUIET_USE, which means
↪  it will use the CRC without complaining. Wait, the code has:

png_set_crc_action(png_handler.png_ptr, PNG_CRC_QUIET_USE, PNG_CRC_QUIET_USE);

According to libpng documentation, PNG_CRC_QUIET_USE means that if the CRC is incorrect, it
↪  will still use the data but not complain. However, if png_crc_finish is called, it will
↪  check the CRC. Wait, in png_handle_sCAL, after reading the chunk data, it calls
↪  png_crc_finish(png_ptr, 0). This checks the CRC of the chunk. If the CRC is invalid, it
↪  will return early, thus not setting the PNG_INFO_sCAL flag.

Therefore, for the first sCAL chunk, the CRC must be correct; otherwise, the code will not
↪  set the valid flag. Similarly, the second sCAL chunk's CRC must be correct to be
↪  processed and trigger the duplicate error.

Calculating CRCs can be complex. However, for the purpose of generating a valid input, we
↪  need to compute the correct CRC for each chunk.

Alternatively, perhaps we can craft the input in such a way that the CRC is correct. This
↪  would require computing the CRC for each chunk.

Let's outline the steps to generate the input:

1. Create the PNG signature: 8 bytes (0x89, 0x50, 0x4E, 0x47, 0x0D, 0x0A, 0x1A, 0x0A).

2. IHDR chunk:
   - Length: 13 (0x00 0x00 0x00 0x0D)
   - Type: 'IHDR' (0x49 0x48 0x44 0x52)
   - Data: For simplicity, set width=1, height=1, bit_depth=8, color_type=2 (RGB),
   ↪  compression=0, filter=0, interlace=0.
   - CRC: Compute over 'IHDR' + data. Need to calculate this.

3. First sCAL chunk:
   - Length: 8 (0x00 0x00 0x00 0x08)
   - Type: 'sCAL' (0x73 0x43 0x41 0x4C)
   - Data: unit=1 (0x01), followed by "1.0\01.0" (hex: 0x31 0x2E 0x30 0x00 0x31 0x2E 0x30)
   - CRC: Compute over 'sCAL' + data.

4. Second sCAL chunk:
   - Length: 4 (0x00 0x00 0x00 0x04)
   - Type: 'sCAL' (0x73 0x43 0x41 0x4C)
   - Data: unit=1 (0x01) followed by three zeros (0x00 0x00 0x00)
   - CRC: Compute over 'sCAL' + data.

5. IDAT chunk:
   - Minimal data, perhaps one byte. But IDAT requires valid zlib-compressed data. However,
   ↪  in fuzzing, maybe the code doesn't process the IDAT data deeply. Alternatively, use a
   ↪  zero-length IDAT chunk.
```

```
6. IEND chunk:
   – Length 0, type 'IEND', CRC.

However, generating valid IDAT data is complicated. But in the provided
↪  LLVMFuzzerTestOneInput, after reading the info, it processes the image data by reading
↪  rows. If the image dimensions are small (1x1), and the IDAT data is valid, this might be
↪  manageable.

Alternatively, perhaps the code can process the IDAT chunk even with invalid data, as the
↪  fuzzer's error handling may allow it. However, the code checks if png_read_row succeeds.
↪  If the IDAT data is invalid, the read might fail, but since the target is to reach the
↪  sCAL handling before that, maybe it's sufficient.

But given the complexity, perhaps the minimal approach is to create a valid PNG with two
↪  sCAL chunks, correct CRCs, and minimal valid IDAT data.

To generate this programmatically, the Python code can construct the PNG bytes step by step,
↪  calculating CRCs where necessary.

For CRC calculation, the zlib library in Python has a crc32 function, which can be used.
↪  Note that PNG uses CRC-32 with the IEEE 802.3 polynomial.

Putting it all together:

– The first sCAL chunk must have valid data and correct CRC.
– The second sCAL chunk must have correct CRC to be processed, but its data can be minimal.

The Python function generate_buf can construct this byte sequence.

The call chain is LLVMFuzzerTestOneInput → libpng's parsing functions (png_read_info, etc.)
↪  → png_handle_sCAL when processing each sCAL chunk.

So the call chain from LLVMFuzzerTestOneInput to the target function would be
↪  LLVMFuzzerTestOneInput → png_read_info → ... (libpng internal functions) →
↪  png_handle_sCAL.

But the user's example in the session used direct function names in the call_chain. Since
↪  the exact path through libpng may involve internal functions, but the key is that
↪  LLVMFuzzerTestOneInput calls into libpng which calls png_handle_sCAL. However, the
↪  user's example in the session included the direct call chain. For the answer, the
↪  call_chain should list LLVMFuzzerTestOneInput leading to png_handle_sCAL via the
↪  necessary functions. However, based on the code provided, the call chain would be
↪  LLVMFuzzerTestOneInput → png_read_info → png_read_info_impl → png_handle_sCAL (assuming
↪  libpng's internal call hierarchy). But without seeing the code of png_read_info, it's
↪  hard to know. However, the user's example included a direct call chain, so perhaps the
↪  call_chain should be LLVMFuzzerTestOneInput → png_read_info → png_handle_sCAL. But since
↪  the user may not have provided all intermediate functions, perhaps the call_chain is
↪  written as ["LLVMFuzzerTestOneInput", "png_read_info", "png_handle_sCAL"].
But according to the code provided, LLVMFuzzerTestOneInput calls png_read_info, which in
↪  libpng processes the chunks. When an sCAL chunk is encountered, png_handle_sCAL is
↪  called. So the call chain is LLVMFuzzerTestOneInput → png_read_info → (libpng internal
↪  processing) → png_handle_sCAL. However, for the purposes of the answer, it's acceptable
↪  to list the direct functions called from LLVMFuzzerTestOneInput leading to the target.
↪  Since the user's code shows that the fuzzer calls png_read_info, which leads to
↪  processing chunks including sCAL, the call_chain would be ["LLVMFuzzerTestOneInput",
↪  "png_read_info", "png_handle_sCAL"].
Thus, the final answer will construct a PNG with two sCAL chunks, ensuring the first is
↪  valid to set the flag and the second triggers the duplicate error.
# Answer:
```json
{
```

```
    "code": "import zlib\n\ndef generate_buf():\n    # PNG signature\n    png_sig =
↪   bytes([0x89, 0x50, 0x4E, 0x47, 0x0D, 0x0A, 0x1A, 0x0A])\n    \n    # IHDR chunk (1x1
↪   RGB)\n    ihdr_data = bytes([\n        0x00, 0x00, 0x00, 0x01,  # width\n
↪   0x00, 0x00, 0x00, 0x01,  # height\n        0x08,  # bit depth\n        0x02,  #
↪   color type (RGB)\n        0x00,  # compression\n        0x00,  # filter\n
↪   0x00   # interlace\n    ])\n    ihdr_type = b'IHDR'\n    ihdr_crc =
↪   zlib.crc32(ihdr_type + ihdr_data).to_bytes(4, 'big')\n    ihdr_chunk = bytes([0x00,
↪   0x00, 0x00, 0x0D]) + ihdr_type + ihdr_data + ihdr_crc\n    \n    # First valid sCAL
↪   chunk\n    scal1_data = bytes([0x01]) + b'1.0\\x001.0'\n    scal1_type = b'sCAL'\n
↪   scal1_crc = zlib.crc32(scal1_type + scal1_data).to_bytes(4, 'big')\n    scal1_chunk
↪   = bytes([0x00, 0x00, 0x00, 0x08]) + scal1_type + scal1_data + scal1_crc\n    #
↪   Second sCAL chunk (duplicate)\n    scal2_data = bytes([0x01, 0x00, 0x00, 0x00])\n
↪   scal2_type = b'sCAL'\n    scal2_crc = zlib.crc32(scal2_type +
↪   scal2_data).to_bytes(4, 'big')\n    scal2_chunk = bytes([0x00, 0x00, 0x00, 0x04]) +
↪   scal2_type + scal2_data + scal2_crc\n    \n    # IDAT chunk (minimal valid zlib data
↪   for 1x1 RGB)\n    idat_data = zlib.compress(b'\\x00\\x00\\x00\\x00\\x00')\n
↪   idat_type = b'IDAT'\n    idat_crc = zlib.crc32(idat_type + idat_data).to_bytes(4,
↪   'big')\n    idat_chunk = len(idat_data).to_bytes(4, 'big') + idat_type + idat_data +
↪   idat_crc\n    \n    # IEND chunk\n    iend_chunk = bytes([0x00, 0x00, 0x00, 0x00]) +
↪   b'IEND' + zlib.crc32(b'IEND').to_bytes(4, 'big')\n    \n    return png_sig +
↪   ihdr_chunk + scal1_chunk + scal2_chunk + idat_chunk + iend_chunk",
    "call_chain": ["LLVMFuzzerTestOneInput", "png_read_info", "png_handle_sCAL"]
}
```

## C CASE STUDIES

In this section, we perform case studies to understand why LLMs fail to solve TTG problems. We analyze the failure reasons and select the relevant response snippets during reasoning to illustrate these issues.

### C.1 FAILURE TYPE 1: HALLUCINATION IN REASONING MODELS

By manually analyzing the responses from LLMs, we found that in many cases, reasoning LLMs tend to hallucinate and rely on fabricated code snippets for decision-making rather than querying the tool (in the agent-based setting). Some of these fabricated code snippets may appear plausible but are not contextually appropriate or reflective of the real codebase, leading to failures in reasoning and incorrect results. Listing C.1 provides an example of this thinking process, where the LLM fabricates a code snippet that does not actually exist and relies on it to reason, ultimately producing a false result.

Listing 13: Example of failure type 1

```
The LLVMFuzzerTestOneInput function would pass the input data to libpng's parsing functions.
↪   The entry point for parsing a PNG file is typically png_read_png or similar. The fuzzer
↪   harness might look like:

 int LLVMFuzzerTestOneInput(const uint8_t* data, size_t size) {
     png_structp png = png_create_read_struct(...);
     png_infop info = png_create_info_struct(png);
     // Set up error handling
     // Read the PNG data
     png_process_data(png, info, data, size);
     // ...
     png_destroy_read_struct(...);
     return 0;
 }

When processing the PNG data, the sCAL chunks are handled by png_handle_sCAL. The first
↪   sCAL chunk is processed normally, setting the info_ptr's valid flag. The second sCAL
↪   chunk triggers the else if (info_ptr != NULL ...) condition, leading to the target
↪   line.
```

## C.2 FAILURE TYPE 2: INABILITY TO SYNTHESIZE VALID INPUT

By manually analyzing the responses from LLMs, we found that although LLMs possess detailed knowledge of valid target file format specifications (as verified through manual prompting), they often struggle to synthesize the correct file format and miss key specifications. Listing C.2 provides an example of LLM-generated code in which it synthesizes an invalid *IHDR* chunk due to neglecting important format specifications. This oversight leads to the generation of incorrect code that fails to meet the required format standards.

**Listing 14: Example of failure type 2**

```python
import zlib

def generate_buf():
    # PNG signature
    png_sig = bytes.fromhex('89504E470D0A1A0A')

    # Helper to create chunks
    def make_chunk(chunk_type, data):
        length = len(data).to_bytes(4, 'big')
        crc = zlib.crc32(chunk_type + data).to_bytes(4, 'big')
        return length + chunk_type + data + crc

    # IHDR chunk (1x1 RGB)
    ihdr_data = bytes.fromhex('00000001 00000001 08020000 0000')
    ihdr = make_chunk(b'IHDR', ihdr_data)

    ...
```

## C.3 FAILURE TYPE 3: FAIL TO REASON PLAUSIBLY

In some cases, we observed that LLMs struggle to make plausible connections between low-level conditions in the code and the higher-level constraints of the target file format. Instead of synthesizing the information and drawing valid inferences, they simply repeat the low-level conditions they encounter, failing to apply reasoning at a higher level. This issue is particularly common in non-reasoning LLMs. For example, in Listing C.3, the LLM simply reiterates the low-level conditions seen in the code, without translating these conditions into meaningful constraints related to the file format. This kind of response highlights the model's inability to reason beyond the surface-level details.

**Listing 15: Example of failure type 3**

```
Based on the code analysis, we can now construct the full call chain and understand the
↪   conditions needed to reach our target line in `png_handle_sCAL`:

Call chain:
1. `LLVMFuzzerTestOneInput`
2. `png_read_info` (via `png_read_png`)
3. `png_push_read_chunk`
4. `png_handle_sCAL`

To reach our target line in `png_handle_sCAL`, we need:
1. A valid PNG header (first 8 bytes)
2. The chunk type must be sCAL (`png_sCAL`)
3. `info_ptr` must not be NULL
4. `(info_ptr->valid & PNG_INFO_sCAL) != 0`
5. `(png_ptr->mode & PNG_HAVE_IHDR) != 0`
6. `(png_ptr->mode & PNG_HAVE_IDAT) == 0`

Here's the input generation code that satisfies these conditions:
...
```

## C.4 Failure Type 4: Missing Constraints in Complex Context

By manually analyzing the responses from LLMs, we found that for complex programs, LLMs sometimes focus narrowly on the target location and surrounding code, overlooking important constraints distributed across the broader context. Interestingly, this is similar to mistakes that humans sometimes make when dealing with complex code, focusing too much on a small part of the program and missing crucial elements that are spread across other sections. For example, Listing C.4 shows a reasoning process where an LLM concludes that the entry function parses an XML file directly from the input buffer. However, upon examining the start of the entry function, shown in Listing C.4, we see that it actually first processes an integer length field followed by the XML file. This oversight demonstrates the LLM's failure to correctly connect and reason about the larger flow of the program, resulting in an incomplete understanding of the code's behavior, which highlights the challenges LLMs face when handling intricate, context-dependent code and emphasizes the importance of maintaining a holistic view of the program during reasoning.

---

**Listing 16: Example of failure type 4**

```
In the LLVMFuzzerTestOneInput function, the fuzzer reads the input data, initializes the
↪  parser, and processes the XML document. The docBuffer comes from xmlFuzzMainEntity,
↪  which is part of the fuzzer's input. So, the input needs to be a valid XML document with
↪  an element that has a prefix but no namespace URI.
```

---

**Listing 17: Entry function in failure type 4 example**

```c
int
LLVMFuzzerTestOneInput(const char *data, size_t size) {
    static const size_t maxChunkSize = 128;
    xmlDocPtr doc;
    xmlParserCtxtPtr ctxt;
    xmlTextReaderPtr reader;
    xmlChar *out;
    const char *docBuffer, *docUrl;
    size_t docSize, consumed, chunkSize;
    int opts, outSize;

    xmlFuzzDataInit(data, size);
    opts = xmlFuzzReadInt();
    opts &= ~XML_PARSE_XINCLUDE;

    xmlFuzzReadEntities();
    docBuffer = xmlFuzzMainEntity(&docSize);
    docUrl = xmlFuzzMainUrl();
    if (docBuffer == NULL)
        goto exit;

    /* Pull parser */

    doc = xmlReadMemory(docBuffer, docSize, docUrl, NULL, opts);

    ...
```

---

# D  Detailed Analysis

## D.1  Number of Queries by LLMs in Agent-Based Setting

We analyze the number of queries emitted by LLMs in the agent-based setting and compare the query counts between reasoning and non-reasoning LLMs. The results, shown in Figure 7, reveal that reasoning LLMs emit significantly fewer queries than their non-reasoning counterparts. We speculate that this may be due to reasoning LLMs being more prone to hallucination.

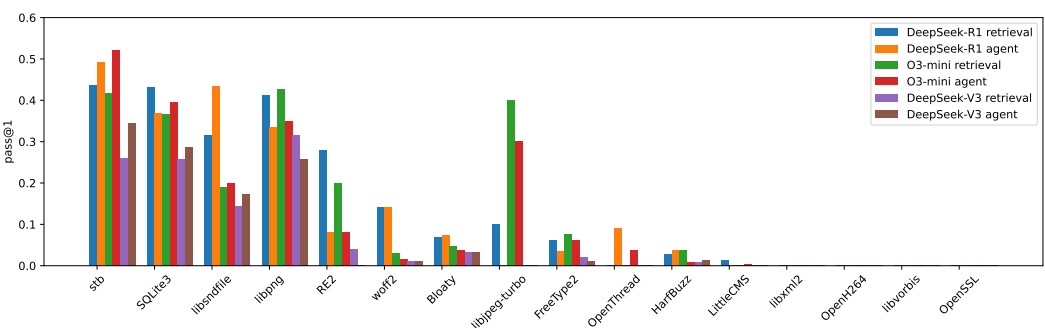

Figure 6: Pass@1 scores across different code repositories.

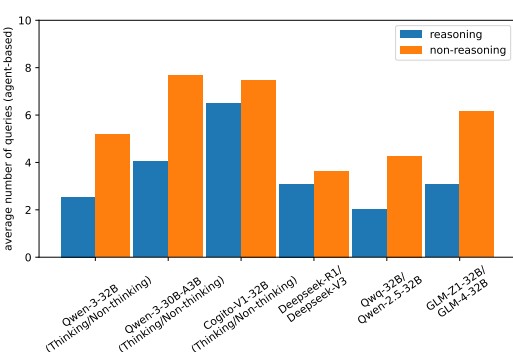

Figure 7: Number of queries emitted by LLMs.

### D.2  PROBLEMS SOVLED BY DIFFERENT LLMS

In this section, we analyze the problems solved by different LLMs across all 500 problems in 5 trials, under both retrieval-based and agent-based settings. Exemplary Venn diagrams of the well-performing models are shown in Figure 8. The upset diagram is displayed in Figure 9, which further highlights the unique problem sets solved by each model. The results demonstrate that even though the overall performance of each model varies, they each manage to solve a distinct subset of problems that the others cannot. This observation underscores the complementary strengths of each model, as different models excel in addressing various aspects of the TTG problem. Such diversity in performance suggests that no single model is universally superior, and each has its own strengths when tackling different parts of the problem space.

## E  LIMITATIONS

The limitations and future directions of this paper include: (1) expanding the evaluation to multiple programming languages to assess LLMs' comprehension across different programming languages, and (2) decomposing the TTG problem into more granular tasks that reflect specific aspects of the comprehensive abilities of LLMs. This will allow for a more detailed understanding of LLMs' strengths and weaknesses in different contexts and domains, and help drive further improvements in their performance across a broader range of software engineering tasks.

## F  LLM USAGE

During the preparation of this manuscript, we made selective use of Large Language Models (LLMs), specifically, as a writing assistant for grammar correction and stylistic refinement. All

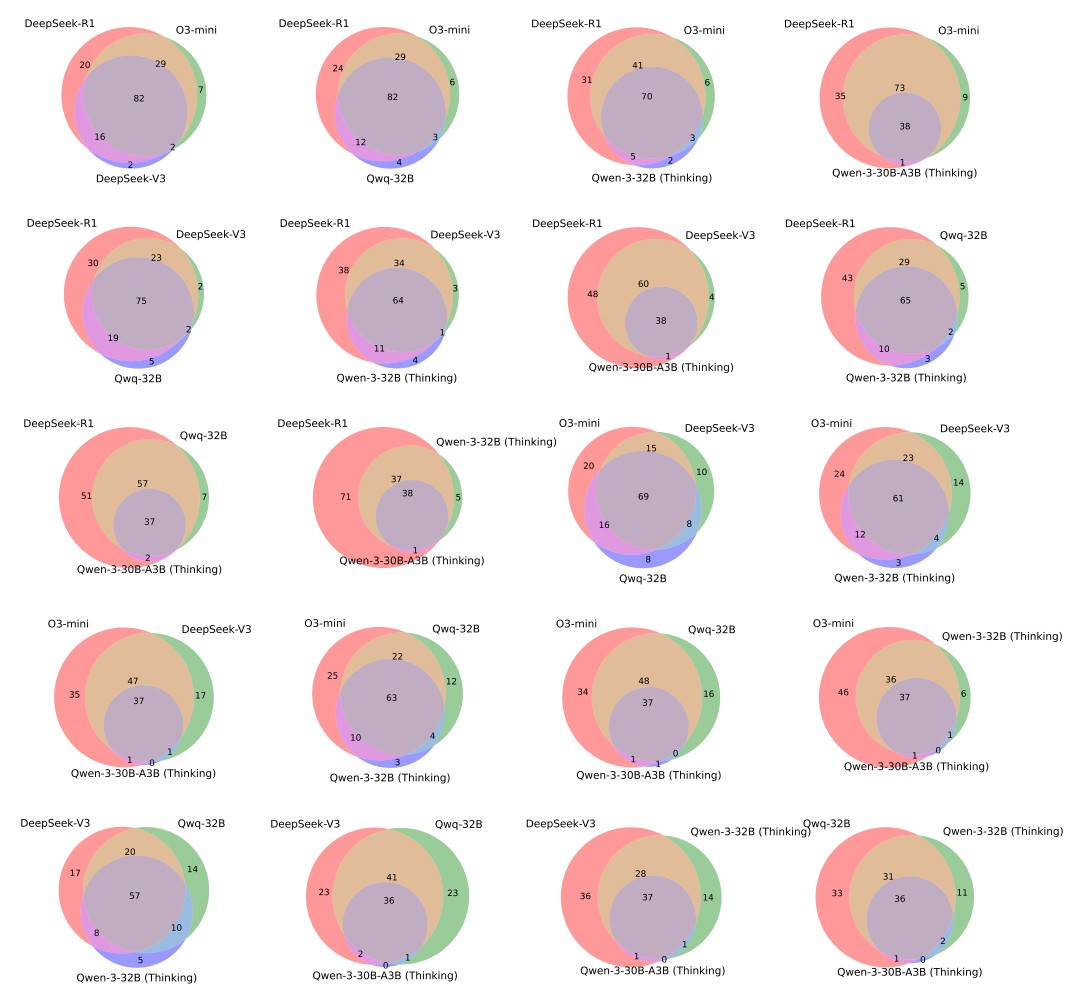

Figure 8: Venn diagrams of solved problems by different LLMs.

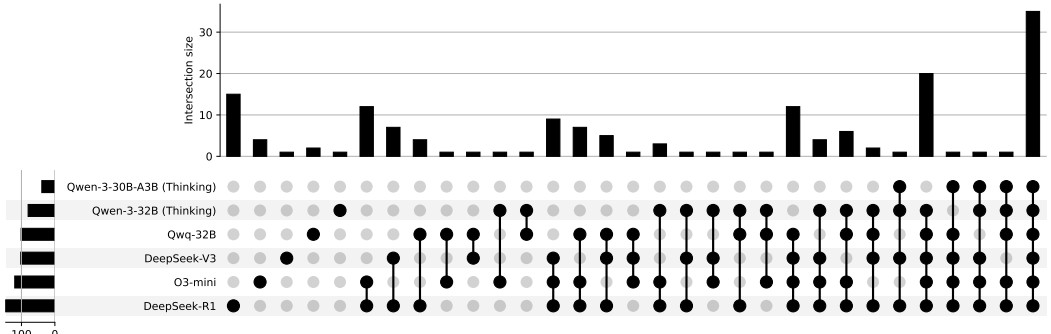

Figure 9: Upset diagram of solved problems by different LLMs.

scientific contributions were conceived and executed entirely by the authors. The LLM did not contribute to any substantive intellectual content.

