# OpenReview forum: "Automated Benchmark Generation for Repository-Level Code Input Synthesis via Coverage-Guided Fuzzing"
_ICLR.cc/2026/Conference — ICLR 2026 Conference Withdrawn Submission_

### Official Review · Reviewer_dxW4 · 2025-10-28

**Soundness:** 2
**Presentation:** 2
**Contribution:** 2
**Rating:** 2
**Confidence:** 4

**Summary:**

This paper presents a novel automated approach for generating a repository-level unit test generation benchmark using coverage-guided library fuzzing. The proposed framework, TTC-Gen, focuses on targeted test-input generation (TTG), where the generated inputs are designed to execute specific targeted parts of the software. Using TTC-Gen, the authors constructed TTC-Bench-Lite, a benchmark consisting of 16 open-source C/C++ projects. They evaluated 18 LLMs on TTC-Bench-Lite and provided an analysis of the effects of reasoning, retrieval-based methods, and agentic approaches.

**Strengths:**

1. **Verified benchmark with oracle**: TTC-Gen utilizes code locations already covered by fuzzing as the oracle. This feature provides soundness in evaluating an LLM’s reasoning ability on code. (Note: soundness in the use of this task for evaluation, not to the method itself.)

2. This approach offers an interesting solution to the contamination problem in LLM evaluation by continuously selecting reachable yet non-trivial code locations.

**Weaknesses:**

1. The authors make several claims about the proposed TTC-Gen approach (Section 3.2); however, their validity is not sufficiently justified.
   1. Feasibility: I agree with the authors on this point.
   2. Non-triviality: A later-discovered location does not necessarily correspond to a deep or hard-to-discover program state. Modern fuzzers (in this paper, LibFuzzer) incorporate seed scheduling, assigning higher energy to input seeds that frequently contribute to coverage. Therefore, *descendants of the same seed* at a later iteration indicate deeper program states. If only time and coverage are measured (as shown in Algorithm 1), it is highly likely that the result corresponds to the beginning of a new and easy-to-cover branch, path, or sub-AST.
   3. Structural requirement: One limitation of fuzzing is that the fuzzer might become stuck in error-handling code instead of exploring the functional logic of the software. Simple coverage-guided gray-box fuzzing cannot rule out these error-handling locations.

2. **Application**. I find the goal of the paper somewhat unclear. Is the paper solely evaluating an LLM’s ability to reason about how to reach a specific program point, or does it provide insights into real-world applications in LLM-based software engineering and testing? I do not think this work offers practical applications, since all targeted locations are already discovered by fuzzing. Moreover, this benchmark does not test the outputs of executions to ensure correct program logic, which is the core goal of unit testing (and other self-assessment testing).
    The authors could argue for *directed testing*, where generating test cases using LLMs is faster than fuzzing (which usually takes days) to cover the same program locations. However, this benchmark assumes that the target location is already known and that we only need to generate a test case to cover it. In this case, I do not see the advantage of this LLM-based approach over IJON [1], where developers can simply annotate the target location to help the fuzzer reach it.

3. **Unclear technical details**. One of TTC-Gen’s main advantages is scalability. However, the authors did not disclose how long each project in the benchmark was fuzzed. This information directly affects how scalable the method is and whether the selected locations are genuinely difficult to find.

4. **False positives**. Whether an input covers a specific program point could be due to side effects. In other words, the LLM might generate inputs that unintentionally execute the target point, even if it was not explicitly reasoning toward it. I suggest that the authors inspect and analyze reasoning summaries or output tokens from the LLMs to verify this.

5. **Presentation**. The authors could improve the presentation in Sections 3 and 4. For example, Section 3.1 relies heavily on subsubsections, whereas Section 4 uses paragraphs that might be better organized as subsubsections. Mixing these structures can cause confusion for readers. In Section 4, as a benchmark paper, the authors could devote more space to discussing the intuition behind the evaluation results rather than only reporting which models perform better. Additionally, the font sizes in the tables and figures are too small to read comfortably.

6. **Related work**. The authors could provide a more comprehensive overview of the current state of LLM-based test generation, and how fuzzing (or other dynamic testing techniques) are used to enhance LLM-based software engineering.

[1]: C. Aschermann, S. Schumilo, A. Abbasi and T. Holz, "Ijon: Exploring Deep State Spaces via Fuzzing," *2020 IEEE Symposium on Security and Privacy (SP)*.

**Questions:**

Please address concerns in weaknesses.

---

### Official Review · Reviewer_f7jB · 2025-10-28

**Soundness:** 2
**Presentation:** 2
**Contribution:** 3
**Rating:** 4
**Confidence:** 4

**Summary:**

This work proposes leveraging fuzzers to design benchmarks for the capability of LLMs at understanding code. Concretely, they propose to use fuzzers to generate inputs, and use those inputs that took a long time to discover (and the respectively new covered line of code) as a reference for model capabilities of reaching exactly those newly discovered lines of code. They implement this proposal on a range of C/C++ repositories and show that SOTA LLMs struggle on generating test inputs that execute the desired line of code, both in a zero-shot setup with code retrieval, as well as in an agentic setup.

**Strengths:**

I think the fundamental idea of leveraging difficult-to-reach lines of code to assess code reasoning is sound and well motivated. I also agree with the authors that this results in a generally well scalable and difficulty-adjustable benchmark. The core results show that this task is indeed difficult even for SOTA LLMs.

**Weaknesses:**

I think there are several easily remediable weaknesses to this approach, all of which need to be addressed thoroughly before the paper is ready for publication.
Most importantly:
- One of the core assumptions or claimed insights of the paper is that lines of code that are found to be executable only very late in a fuzzer execution are more difficult to create test inputs for (Line 237, Line 84, ...). This makes intuitive sense, but the paper lacks experimental evidence to back this claim. Strong evidence would comprise collecting many different lines of code at different discovery points and evaluating 1-2 models on them, showing that model success rate is generally higher for earlier discovered lines.
- The evaluation in an agentic setting introduces its own agent with specific tools. This is justified with the claim that standard agents are not designed for the task of fuzzing. A comparison to established, SOTA code agents (OpenHands [1], SWE-Agent [2]) would help confirm that this is indeed the case. Ideally, I would like to further see the authors supply OpenHands and SWE-Agent (or similar agents) _additionally_ with their proposed tools to establish that these indeed increase performance on the task. Currently it is hard to estimate whether models indeed fundamentally struggle on the given task or whether the evaluation harness is simply making the task unnecessarily difficult. One qualm I have with the current setup for example is that it seems the model is not actually to able to execute generated code, making it much more difficult to adequately refine their generated tests.
- The evaluation asks LLMs to generate scripts that generate inputs to their target functions. This is justified by the claim that models struggle at directly generating the relevant bytes. A small experiment to validate this claim would provide evidence for this claim is missing, for example, evaluating the best LLM in the code generation setting on generating bytes directly.

Several further issues:
- The design of the benchmarks prohibits evaluating LLMs that outperform fuzzing libraries (in the future). This might be worth discussion.
- Key statistics about the benchmark are missing. How many instances are present for each library / on average per library? What are typical expected file input sizes? How many distinct files/functions are targeted in total / per repository? I may have forgotten about other typical statistics.
- Somehow the demonstration for solving the task in the user prompt (listing 2) has a different wording than the user prompt (listing 3). This might be confusing to the model and seems unusual to me.
- There are two related works about LLMs used as general purpose fuzzers, also on repository level code which are not mentioned. For example [3,4]
- The presentation requires major work in several locations
  - Line 288 - 305 starts with a discussion of limitations of prior work / advantages of their work in the middle of the method section. I think this would be better placed in the introduction or related work.
  - Table 3 is very hard to read. Instead of providing a sparsely populated column for reasoning mode and whether it is enabled, I suggest seperating the table horizontically into top (reasoning enabled) and bottom (non-reasoning or reasoning not enabled).
  - Table 3 should be sorted by performance or name or size or anything really. Alterantively/additionally highlight the best model in each column in boldface
  - Figure 4/5 are too small and badly formatted. The y axis should show % instead of 0.X, the legend should contain capitalized words and the figures should span the text width.
  - The captions of table 3 and figure 4/5 are non descriptory. They should contain key details about the depicted data and some key insights to be drawn from the data.
  - Table 2 would benefit from more vertical space between rows and grouping by input file types (images/audio/video/other/...)
  - Figure 8 is not legible. Maybe it would be enough to show per model overlaps in agent/retrieval mode? Or overlaps between top 3-5 models only.

Nitpicks:
- Typo in L 102 (generate)

[1] Wang et al, OpenHands: An Open Platform for AI Software Developers as Generalist Agents, ICLR 2025
[2] Yang et al, SWE-agent: Agent-Computer Interfaces Enable Automated Software Engineering, NeurIPS 2024
[3] Deng et al, Large Language Models Are Zero-Shot Fuzzers: Fuzzing Deep-Learning Libraries via Large Language Models, 2023
[4] Zhang et al, How Effective Are They? Exploring Large Language Model Based Fuzz Driver Generation, 2024

**Questions:**

Please see the questions in the weaknesses section.

---

### Official Review · Reviewer_gs7S · 2025-10-30

**Soundness:** 3
**Presentation:** 3
**Contribution:** 3
**Rating:** 4
**Confidence:** 3

**Summary:**

This paper introduces TTH-GEN, an automated framework for generating repository-level code comprehension tasks, and the resulting benchmark TTG-BENCH-LITE. The benchmark evaluates whether large language models (LLMs) can understand real-world C/C++ programs and synthesize inputs that trigger specific execution paths. Using coverage-guided fuzzing (CFG), the authors construct 500 verified tasks across 16 open-source projects. Experimental results show that even the strongest LLMs achieve at most 14.6% pass@1, while reasoning-optimized models consistently outperform standard ones.

**Strengths:**

1. The paper builds TTG-BENCH-LITE, an automated dataset that evaluates repository-level code comprehension and input synthesis abilities of LLMs, going beyond function or file level benchmarks.

2. Instead of using easily reachable code locations discovered through random fuzzing, the benchmark focuses on non-trivial, structure-aware targets that require reasoning over complex control and data flow to reach.

3. Unlike previous benchmarks that depend on human-authored documentation or unit tests, the proposed TTG-GEN framework uses Coverage-Guided Fuzzing (CGF) to automatically and programmatically generate benchmark problems at scale.

4. The method leverages the CGF concept of interesting inputs and applies a discovery-time threshold to exclude trivially reachable locations, ensuring that only non-trivial, meaningful targets are included in the dataset.

5. The authors conduct extensive experiments across a wide range of LLMs under both retrieval-based and agent-based settings, providing a broad and insightful comparison.

**Weaknesses:**

1. The benchmark uses only 16 C/C++ repositories, which seems rather limited for capturing the diversity of real-world software projects. Moreover, the selection criteria for these repositories are not clearly explained. For instance, whether they are chosen based on size, complexity, or domain. Given that model performance varies significantly across repositories, the absence of a clear selection rationale undermines the representativeness of the benchmark.

2. Although the benchmark effectively measures code reachability and structural reasoning through CGF, it remains unclear whether these automatically generated targets correspond to semantically meaningful comprehension of repository-level code. Since the tasks are defined purely based on coverage novelty rather than real functionality or behavioral semantics, it is uncertain whether high performance truly reflects deeper code understanding.

3. While the paper evaluates a wide variety of models, most of the analysis remains quantitative and surface-level. There is limited qualitative insight into why certain models perform better or fail, or what kinds of reasoning strategies they adopt.

4. The paper does not include any evaluation of the benchmark’s intrinsic quality. For example, by comparing machine-generated tasks against human-created ones or by validating tasks meaningfulness through human judgment. Without such assessment, it remains unclear whether TTG-BENCH-LITE reliably measures the intended reasoning or comprehension capabilities of LLMs.

**Questions:**

1. In section 4.2.1, the authors attribute the performance gains of reasoning models entirely to reinforcement learning. Could the author clarify whether any ablation or causal analysis was conducted to isolate the effect of reinforcement learning? If not, how confident are the authors that reinforcement learning is the primary factor behind the observed improvements?

2. In Figure 4, could the authors specify whether the main text explains what the left and right sides correspond to? The distinction seems unclear from the written description. In addition, the current labels in the figure are quite small and hard to read, making it difficult to identify which side corresponds to the retrieval-based setting and which to the agent-based setting.

---

### Official Review · Reviewer_zbiB · 2025-11-01

**Soundness:** 2
**Presentation:** 3
**Contribution:** 3
**Rating:** 6
**Confidence:** 4

**Summary:**

The paper introduces TTG-GEN, a framework for generating targeted test-input generation (TTG) problems from real-world codebases to assess the capability of LLMs to synthesize input byte sequences to execute specific, designated code locations. The evaluation shows that state-of-the-art models exhibit critically low success rates, a indicating these tasks are a significant hurdle for current models and highlighting considerable room for improvement in LLMs.

**Strengths:**

**Originality**
The paper introduced the first large-scale benchmark from real-world GitHub PRs to evaluate the proactive defect discovery capabilities of LLMs at the repository level.

**Quality**
The paper made extensive experiments on various LLMs and two different settings for handling repo-level codebase, showing quantitative results of with pass@1-5 scores.

**Clarity**
The paper is well-written with description of benchmark construction workflow and evaluation process. The appendix provides detailed prompt.

**Significance**
The paper proposed a large-scale benchmark which can help researchers in AI and SE fields to measure LLM's capability to generate targeted test input aiming to reach specific code location.

**Weaknesses:**

**Coarse-Grained Metric**
The paper uses pass@1-5 scores to measure the performance of LLM-generated input to reach target location. However, it does not consider the fine-grained quality of inputs in satisfying the multiple constraints to reach target location. Some inputs may be very close to reaching target location while others may fail at very beginning. The authors may consider use fine-grained evaluation metrics to measure the distance of the input to the target location.

**Correlation between task difficulty for fuzzing and LLM**
The paper uses heuristic threshold of fuzzing time (one hour) to construct benchmark of non-trivial target locations. The assumption behind this is that the difficulty for fuzzing to reaching certain location is correlated with the difficulty for LLM to generate targeted test input. But it's difficult for mutation-based fuzzing to bypass certain constraints like magic numbers, while LLMs are expected to be good at reasoning on high-level constraints. It would be interesting to see results about correlation between fuzzing time and LLM's performance, which would also help understand the quality of the benchmark.

**Questions:**

1. What's the distribution of fuzzing time needed to reach the target locations in the benchmark?
2. Do LLMs perform better on text-like format than binary format?
3. Do LLMs rely on format-specific third-party library to generate input for some non-trivial format?

---

### Note · Authors · 2026-01-26

**Comment:**

We thank the reviewers for their time and constructive feedback. All reviewers recognize that our work presents a scalable and contamination-resistant approach for evaluating LLMs on real-world repository-level targeted test-input generation. We will incorporate their suggestions to improve this paper.

**Withdrawal Confirmation:**

I have read and agree with the venue's withdrawal policy on behalf of myself and my co-authors.

---

### Meta-Review · Area_Chair_f6GW · 2026-01-06

**Summary:**

Most of the reviewers gave leaning-reject review comments and one of the reviewer marked above threshold, while the authors did not participate in the discussion. So the concerns from all the reviewers are still outstanding. The AC agree with the reviewers to reject the paper.

**Reviewer Concerns:**

Reviewers have concerns that the assumption: the difficulty for fuzzing to reaching certain location is correlated with the difficulty for LLM to generate targeted test input, does not have enough evidence support. They also show concerns on the evaluation, benchmark, and fairness of the comparison. Besides that, the writing also needs improvement.

**Reviewer Scores:**

Won't change.

---

### Decision · Program_Chairs · 2026-01-26

Reject